# When Visualizing is the First Step to Reasoning: MIRA, a Benchmark for Visual Chain-of-Thought

## Abstract

We propose **MIRA** (**M**ultimodal **I**magination for **R**easoning **A**ssessment), a new benchmark designed to evaluate models in scenarios where generating intermediate visual images is essential for successful reasoning. Unlike traditional Chain-of-thought (CoT) methods that rely solely on text, tasks in **MIRA** require models to generate and utilize intermediate images — such as sketches, structural diagrams, or path drawings — to guide their reasoning process. This setup closely mirrors how humans solve complex problems through "drawing to think". To solve this, **MIRA** focuses on tasks that are intrinsically challenging and involve complex structures, spatial relationships, or reasoning steps that are difficult to express through language alone (*e.g.*, tracking a die's movement on a board and summing the face-down values after each roll). To ensure that our evaluation data is of high-quality, we include 546 multimodal problems, annotated with intermediate visual images and final answers. We also propose a unified evaluation protocol for **MIRA** that spans three levels of evaluation input: direct input with image and question only, text-only CoT (Text-CoT) input with image and thinking prompts, and Visual-CoT input with both annotated image clues and textual thinking prompts. To probe the upper bound of model capacity on our benchmark, we also report pass@$k$ and majority voting accuracies under different $k$ settings. Experimental results show that existing multimodal large language models (MLLMs), including strongest private models (*e.g.*, GPT-5, o3, Gemini 2.5 Pro) as well as strong open-weight models (*e.g.*, Qwen2.5-VL, GLM 4.5V), perform poorly when relying solely on textual prompts. However, when intermediate visual cues are provided, model performance improves consistently, yielding an average relative gain of 33.7% across all models and tasks. We also probe the upper bound by expanding the search space and designing textual prompts aligned with Visual-CoT, but both yield only limited improvements compared to our Visual-CoT setting. These results underscore the critical role of imagined visual information in enabling successful reasoning on **MIRA**.

## 1 Introduction

Chain-of-Thought (CoT) prompting has emerged as a powerful paradigm for improving the reasoning capabilities of large language models (LLMs) (Wei et al., 2022). By generating intermediate natural language rationales, CoT enables models to decompose complex problems into manageable steps, yielding significant gains in tasks such as arithmetic reasoning, commonsense inference, and multi-hop question answering (Kojima et al., 2022; Zhang et al., 2022). Despite its effectiveness, existing CoT methods operate entirely in the textual domain—even for multimodal models: every intermediate step must be verbalized in words. This purely linguistic format is inherently limiting, as many real-world reasoning problems are intrinsically visual — requiring spatial reasoning, geometric manipulation, or physical simulation — that humans typically address by drawing to think. In such cases, natural language sometimes becomes an awkward and lossy medium for expressing intermediate states, forcing models to describe visual cues step-by-step. As AI models showing exceptionally strong capabilities on everyday tasks with resemblant human perceptions, questions arise when facing these real-

world questions: *Can current multimodal models truly reason with integrated visual artifacts*, and *how much can this capability contribute to solving complex visual reasoning problems?*

Existing multimodal reasoning benchmarks primarily treat individual images as the input, testing models on tasks such as visual question answering (Fu et al., 2023; Antol et al., 2015; Yue et al., 2024a;b; Yu et al., 2023; Phan et al., 2025), image captioning (Lin et al., 2014; Dong et al., 2024; Cheng et al., 2025a), or visual grounding (Yu et al., 2016; Mao et al., 2016). While some recent datasets incorporate multi-step reasoning (Lu et al., 2022; Wang et al., 2025; Chen et al., 2025c; Cheng et al., 2025b; Wu et al., 2025b), the intermediate steps remain text-only, and visual generation is rarely required to solve these problems. A few preliminary efforts have explored tool-augmented reasoning, where models can call external drawing components or retrieve related images (Hu et al., 2024; Zhang et al., 2024; Fu et al., 2025; Mallis et al., 2024; Shen et al., 2024), but these are often limited to specific domains and constrained by tools leveraged.

To bridge this gap, we introduce **MIRA** (**M**ultimodal **I**magination for **R**easoning **A**ssessment), a benchmark designed to evaluate reasoning scenarios where generating or leveraging intermediate visual representations is essential. Each instance is constructed according to three principles: (1) requiring intermediate visual cues to answer the question, (2) pairing each instance with annotated step-wise visual clues to enable evaluation under a Visual-CoT setup, and (3) enforcing strict human annotation and cross-validation to guarantee data quality. **MIRA** includes both tasks that hinge on a single auxiliary image and those requiring a sequence of intermediate visual states (e.g., tracking object state changes over time). In total, the benchmark spans 20 task types and 546 carefully designed examples, covering scenarios from spatial layout reasoning and geometric construction to cross-temporal state tracking. All examples are paired with gold-standard intermediate visual states and images precisely aligned with reference reasoning trajectories, along with final answers, ensuring evaluation is automated and repeatable.

Since each problem is paired with human-annotated intermediate visual states, to explore the capacity of models given different granularities of visual information, we evaluate models under three settings: (1) Direct Evaluation — giving question and image directly; (2) Text-CoT Reasoning — giving CoT prompt with the question and image; and (3) Simulated Visual-CoT Reasoning — giving both visual step input and textual CoT prompt along with the question and image. This protocol decouples the information contribution of visuals from textual generation ability and provides an evaluation path for future MLLMs that can "think while drawing". We select state-of-the-art open-weight MLLMs and commercial MLLMs from six different companies. By employing these three input settings, our analysis uncovers several key findings. **MIRA** proves highly challenging: even GPT-5 reaches only 16.5% accuracy with direct inputs, and no model surpasses 20%. Some categories are particularly difficult, such as Puzzles (9.5% *vs.* 16.1% on others). Text-CoT, while useful elsewhere, often underperforms here on **MIRA** reducing accuracy for Gemini 2.5 Pro and o3 by 18.3% and 14.0%. In contrast, our Visual-CoT delivers consistent gains, *e.g.*, GPT-5-mini improves from 13.7% to 23.2% on average, and Physics tasks nearly double across all proprietary MLLMs. Together, these results highlight both the limits of text-only CoT prompting and the promise of visual reasoning for existing advanced multimodal systems.

## 2 RELATED WORK

**CoT Reasoning in LLMs.** Prompting models to articulate step-by-step solutions in natural language - i.e., chain-of-thought prompting - significantly improves their reasoning performance (Wei et al., 2022). Building on this paradigm, variants like zero-shot CoT (Kojima et al., 2022) and automatically constructed CoT exemplars (Zhang et al., 2022) enable models to break down complex problems into intermediate textual steps, achieving strong results on arithmetic, commonsense, and multi-hop QA tasks. However, these approaches remain purely textual: they assume verbal reasoning alone suffices and struggle on inherently visual tasks that are better served by diagrams or spatial representations, where intermediate graphical states would be needed.

**Reasoning Benchmarks in MLLMs.** Multimodal reasoning research has been driven by benchmarks like Visual Question Answering (VQA) (Antol et al., 2015; Fu et al., 2023; Yue et al., 2024a), image captioning (Lin et al., 2014; Dong et al., 2024), and visual grounding (Yu et al., 2016; Mao et al., 2016). These tasks use an image-in, text-out format focusing on visual understand-

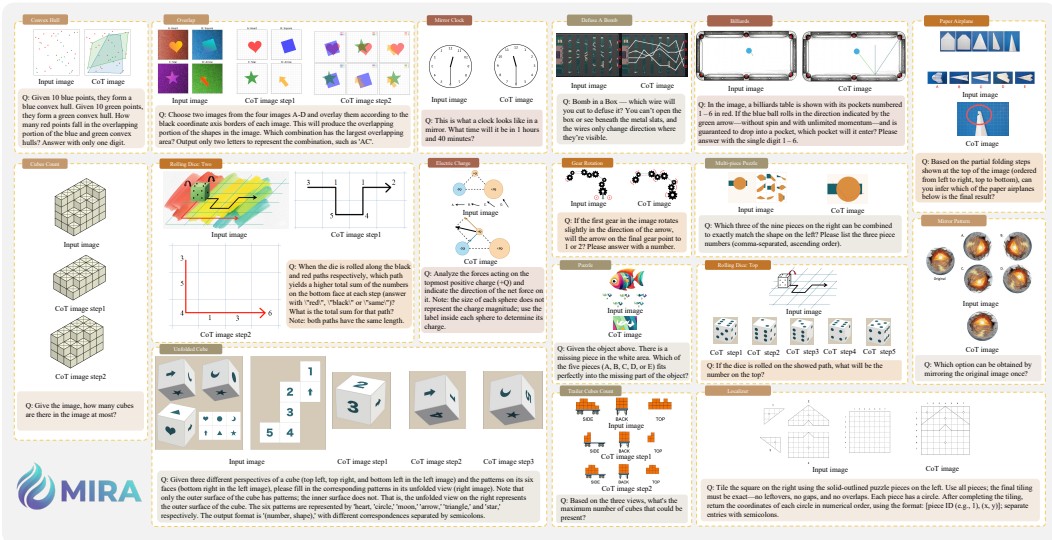

Figure 1: **MIRA** categorizes Visual-CoT reasoning tasks into two primary types: Static (Single-Step) and Dynamic (Multi-Step), with representative examples from each category illustrated in the figure. The dataset includes 20 types of tasks, 546 input images with manually designed questions, and 936 manually constructed single-step and multi-step intermediate images. For more cases, please refer to Appendix D.

ing, and many prompts target simple perception (e.g., "Who is wearing glasses?"), placing minimal demands on reasoning. Some datasets, such as ScienceQA (Lu et al., 2022), include multi-step reasoning hints, but those steps remain textual; problems can still be solved via natural-language rationales alone, without requiring intermediate images or visual cues. Comprehensive evaluation suites broaden the coverage of vision–language tasks but similarly do not require models to generate or manipulate intermediate visual cues (Phan et al., 2025; Yu et al., 2023; Yue et al., 2024a;b; Fu et al., 2023). Thus, while these benchmarks advanced MLLM evaluation, they overlook problems that genuinely need intermediate visual or diagrammatic reasoning — capabilities vital for unified generation - understanding models (e.g., (Deng et al., 2025; Chen et al., 2025a;b; Xie et al., 2024; Wu et al., 2025a; Cai et al., 2025)).

**Integrating Visual Cues into Reasoning.** Bridging the gap between human problem solving (which often uses sketches or diagrams) and text-only model reasoning, recent work explores visual chain-of-thought techniques. For example, Visual CoT (Shao et al., 2024) augments textual rationales with intermediate visual cues (e.g., bounding boxes on relevant regions) to improve vision–language reasoning. Beyond static cues, tool-augmented methods like Visual ChatGPT (Wu et al., 2023), VisProg (Gupta & Kembhavi, 2023), ViperGPT (Surís et al., 2023) allow models to call external drawing or vision tools during reasoning to produce auxiliary visuals (sketches, cropped views, highlighted regions). Further, frameworks such as Vision-Augmented Prompting (Xiao et al., 2024) and Visual Sketchpad (Hu et al., 2024) let models execute code (e.g., Python) to generate or update diagrams that assist in solving geometry and spatial reasoning tasks. However, these approaches rely on external tool orchestration and have not yet been systematically evaluated in open-ended reasoning scenarios.

**Unified MLLMs with Image Generation Capabilities.** Recent unified-architecture MLLMs (e.g., Blip3-o (Chen et al., 2025a), Janus-pro (Chen et al., 2025b), Bagel (Deng et al., 2025), Show-o (Xie et al., 2024), OmniGen2 (Wu et al., 2025a)) combine vision and language processing to both understand and generate images. By contrast, some open-weight MLLMs (Qwen2.5-VL (Bai et al., 2025), InternVL-X3 (Zhu et al., 2025)) focus only on visual understanding (e.g., recognition, grounding, VQA) and do not support general image generation. In principle, generation-capable models could produce intermediate sketches or diagrams during reasoning, akin to a human's scratch paper. Yet most image-generating MLLMs are optimized for photorealistic synthesis or descriptive captioning, not for creating abstract, task-specific diagrams; even advanced systems like Gemini (Comanici et al., 2025b) and GPT-5 (Hurst et al., 2024) have not demonstrated robust

"think-while-drawing" abilities. This gap highlights the need for benchmarks explicitly evaluating a model's ability to generate and use intermediate visual representations during reasoning precisely what our proposed **MIRA** benchmark is designed to assess.

# 3 MIRA: MULTIMODAL IMAGINATION FOR REASONING ASSESSMENT

In this section, we introduce **MIRA** a comprehensive benchmark designed to evaluate capacities of MLLMs for Visual-CoT reasoning across a broad scope of tasks. **MIRA** consists of 546 curated samples spanning four challenging domains: Euclidean Geometry (EG), Physics-Based Reasoning (PBR), Abstract Spatial & Logical Puzzles (ASLP), and Causal Transformations (CT). Each instance is meticulously designed through a pipeline involving extensive human annotations to ensure high quality and a unique ground-truth answer. Except the data content itself, our evaluation extends beyond the vanilla direct input evaluation by requiring models to engage in complex, multi-step visual reasoning, which is further analyzed through a novel three-level diagnostic protocol with provided sequences of visual clues.

## 3.1 BENCHMARK DATA DESIGN AND CONSTRUCTION DETAILS

The design of **MIRA** data is built around three core principles to ensure the data requires visual-CoT to solve, paired with artificial visual reasoning clues, and is of high-quality, respectively. *First*, our data design emphasizes the need of intermediate visual information (*i.e.*, Visual-CoT) to solve questions. This intermediate process is analogous to the scratchpad diagrams humans create when solving difficult problems. For instance, to determine the direction of force on a positive charge, one might draw a force-body diagram to visualize the net force. This approach is a complement to traditional text-based CoT and other prompting paradigms that simulate model thinking merely as attention-grounding bounding boxes or textual descriptions of visual concepts.

*Second*, to probe the capacities of models to process visual CoT information, we parameterize the complexity of our data by the number of reasoning steps and intermediate visuals. Specifically, **MIRA** evaluates this capability across either single-step or multi-step visual reasoning - requiring either one pivotal intermediate visual clue or a sequential of visual trajectories during model inference.

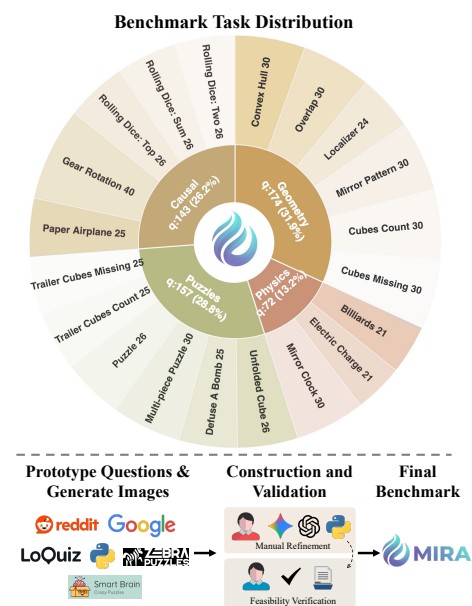

Figure 2: A high-level overview of the **MIRA** data design and construction pipeline.

*Third*, diving into detailed data implementation, we employed a hybrid construction pipeline by integrating the manual labeling, human inspection and programmatic generations. All manually created problems were authored by graduate-level researchers, drawing inspiration from Reddit's visual-puzzle and puzzle-game communities, as well as various exercise repositories and brain-teaser websites (Reddit, 2025; Braingle, 2025; Smart Brain, 2020; LoQuiz, 2025), while ensuring novel formulations and original content. To complement these, additional problems were programmatically generated via Python scripts, enabling fine-grained control over difficulty. These initial image inputs are then refined for better visual quality and clarity using image editing tools (*e.g.*, GPT-4o, Gemini 2.5 Flash). The final stage involves a rigorous quality control with cross-review and conflict resolve, to ensure each problem has a single, unambiguous ground-truth answer and a reliable visual reasoning trajectory for the input. Figure 2 summarizes the detailed data pipeline.

## 3.2 EVALUATION PROTOCOL

Visual-CoT tasks are difficult in nature, as they favor intermediate visual clues for answering faithfully. A key contribution of **MIRA** is its three-level diagnostic evaluation protocol, designed to move beyond a single accuracy score and provide insights into why a model fails:

- **Level 1: Direct Evaluation.** The standard end-to-end setting where the model receives only the initial problem $(I_q, T_q)$ and must produce the final answer directly, where the $I_q$ and $T_q$ mean the input images and text, respectively. This measures overall problem-solving capability.
- **Level 2: Text-CoT Reasoning.** The model is prompted to first generate a textual chain of thought and then provide the final answer. This level tests the model's ability to solve problems in **MIRA** using text-based reasoning.
- **Level 3: Simulated Visual-CoT Reasoning.** Considering that current models, both open weight and commercial, cannot accurately generate or interleave the use of intermediate images and tool-generated auxiliary visuals, we provide manually annotated intermediate images for every task in **MIRA**. In Level 3, we evaluate the benefits of these intermediate visuals by providing them to the model and prompting it to reason based on them before giving the final answer.

Together, this evaluation paradigm allows us to assess the model abilities to understand and think over visual-reasoning-intensive situations. We provide detailed prompt templates in the evaluation process for these three levels in Appendix B.

## 4 EVALUATION

This chapter aims to provide a comprehensive empirical evaluation of the performance of current mainstream MLLMs on the **MIRA** benchmark. We have three core objectives: (1) to quantify the performance of state-of-the-art models on tasks that require intermediate visual reasoning; (2) to systematically measure the performance improvements brought by an explicit visual chain-of-thought, thereby isolating and evaluating the actual contribution of visual cues; and (3) to conduct a fine-grained analysis of the models' various capabilities and failure modes, providing deep insights into the key challenges in achieving a human-like "thinking by drawing" reasoning process and how to overcome them.

### 4.1 EXPERIMENTAL SETUP

This subsection provides a detailed account of the methodological framework for our evaluation, covering model selection, implementation of the diagnostic evaluation protocol, and the evaluation metrics, to ensure that our results are reproducible and clear.

**Baseline Models.** To provide a comprehensive snapshot of the current landscape, we selected a diverse and representative cohort of MLLMs and these models are grouped into three key categories.

- **Closed-Source MLLMs:** These models represent the pinnacle of multimodal capabilities and serve as an upper-bound reference. We evaluate a range of leading models, including GPT-5 and GPT-5-mini (OpenAI, 2025a), GPT-4.1 (Fachada et al., 2025), GPT-4.1-mini (Fachada et al., 2025), GPT-4o (OpenAI, 2024b), GPT-4o-mini (OpenAI, 2024a), Claude 4 Opus (Anthropic, 2025), Claude 4 Sonnet (Anthropic, 2025), o4-mini (OpenAI, 2025b), o3 (OpenAI, 2025b), Seed1.5-VL (Guo et al., 2025), Seed1.6 Vision Pro (Guo et al., 2025), Qwen-VL-Max (Bai et al., 2025) and Gemini 2.5 Flash and Pro (Comanici et al., 2025a).
- **Open-Weight MLLMs (Understanding):** This class of models exhibits strong visual understanding capabilities, but typically lacks native, general-purpose image generation abilities. Considering the overall difficulty of the task, we only selected flagship models with large parameter counts from open-weight models for evaluation. We evaluate s Qwen2.5-VL (73B) (Bai et al., 2025), and GLM 4.5 V (106B) (Hong et al., 2025). This category helps us assess the reasoning limitations of models that are primarily geared towards perception tasks.
- **Open-Weight Unified MLLMs (Understanding & Generation):** This emerging class of models possesses both understanding and generation capabilities, making them the most

Table 1: Main results of various models on **MIRA**. The models are grouped into three categories: Closed-Source SOTA MLLMs, Open-Weight MLLMs, and Open-Weight Unified MLLMs. We report model results under three different inputs: **D** for direct input, **T** for Text-Cot, and **V** for Visual-CoT. Detailed results on each sub-category can be found on Tables 4-10. We highlight the top-three performing models in each column with varying shades of blue, where a darker shade indicates a higher rank.

| Model | EG (Geometry) | | | PBR (Physics) | | | ASLP (Puzzles) | | | CT (Causal) | | | Overall | | |
|---|---|---|---|---|---|---|---|---|---|---|---|---|---|---|---|
| | D | T | V | D | T | V | D | T | V | D | T | V | D | T | V |
| *Closed-Source SOTA MLLMs* | | | | | | | | | | | | | | | |
| Gemini 2.5 Flash | 9.4 | 11.7 | 15.6 | 19.7 | 22.9 | 46.7 | 6.5 | 5.9 | 7.1 | 14.0 | 12.0 | 14.1 | 11.3 | 11.7 | 17.3 |
| Gemini 2.5 Pro | 10.6 | 11.1 | 15.0 | 41.1 | 27.1 | 59.5 | 11.0 | 7.1 | 9.7 | 17.2 | 17.0 | 10.1 | 16.9 | 13.8 | 18.9 |
| GPT-5 | 14.5 | 14.4 | 15.6 | 29.9 | 22.2 | 53.7 | 10.8 | 15.7 | 19.9 | 17.9 | 19.3 | 28.6 | 16.5 | 17.2 | 25.9 |
| GPT-5-mini | 10.0 | 10.6 | 20.0 | 28.1 | 21.3 | 39.8 | 7.2 | 10.8 | 16.9 | 17.2 | 13.1 | 24.6 | 13.7 | 12.9 | 23.2 |
| GPT-4.1 | 16.1 | 17.8 | 16.7 | 12.2 | 16.5 | 39.4 | 6.6 | 7.9 | 10.5 | 13.2 | 17.9 | 15.3 | 11.9 | 14.7 | 17.9 |
| GPT-4.1-mini | 5.5 | 8.9 | 11.7 | 9.5 | 22.2 | 31.1 | 12.4 | 12.5 | 10.9 | 10.3 | 15.4 | 14.8 | 9.4 | 13.6 | 15.1 |
| GPT-4o | 17.2 | 11.1 | 11.7 | 8.0 | 11.1 | 38.1 | 4.6 | 3.2 | 9.7 | 14.1 | 12.1 | 9.1 | 11.2 | 9.0 | 14.4 |
| GPT-4o-mini | 7.2 | 13.9 | 11.1 | 14.3 | 5.9 | 17.5 | 7.8 | 6.6 | 9.2 | 15.6 | 17.3 | 15.2 | 10.5 | 11.3 | 12.5 |
| o3 | 15.2 | 13.3 | 18.3 | 22.4 | 16.9 | 47.6 | 11.5 | 8.5 | 12.9 | 20.1 | 20.2 | 27.5 | 16.4 | 14.1 | 23.4 |
| o4-mini | 14.0 | 13.1 | 14.0 | 18.8 | 30.5 | 44.0 | 14.6 | 11.4 | 11.7 | 16.6 | 14.4 | 24.4 | 15.6 | 15.5 | 20.4 |
| Claude 4 Opus | 12.8 | 15.6 | 15.0 | 19.0 | 22.2 | 28.6 | 7.8 | 7.8 | 10.5 | 12.7 | 11.6 | 12.1 | 12.2 | 13.3 | 14.9 |
| Claude 4 Sonnet | 12.2 | 10.0 | 15.0 | 19.7 | 18.6 | 27.6 | 10.3 | 11.0 | 8.5 | 12.6 | 15.1 | 9.6 | 12.9 | 12.9 | 13.6 |
| Seed1.5-VL | 11.1 | 10.6 | 16.1 | 20.6 | 28.6 | 43.7 | 8.6 | 11.2 | 3.9 | 14.0 | 18.0 | 12.6 | 12.5 | 15.3 | 15.7 |
| Seed1.6 Vision Pro | 13.3 | 11.1 | 21.7 | 20.7 | 22.2 | 51.6 | 8.6 | 8.5 | 4.6 | 16.9 | 10.2 | 11.2 | 13.9 | 11.8 | 18.4 |
| Qwen-VL-Max | 11.7 | 12.8 | 17.8 | 24.5 | 22.2 | 31.7 | 13.5 | 9.1 | 11.7 | 13.8 | 7.6 | 20.2 | 14.7 | 11.8 | 18.7 |
| Average | 12.1 | 12.4 | 15.7 | 20.6 | 20.7 | 40.0 | 9.5 | 9.1 | 10.5 | 15.1 | 14.7 | 16.6 | 13.3 | 13.3 | 18.0 |
| *Open-Weight MLLMs (Understanding)* | | | | | | | | | | | | | | | |
| Qwen2.5-VL (32B) | 4.4 | 3.9 | 5.6 | 4.8 | 6.4 | 4.3 | 1.3 | 3.9 | 4.5 | 3.9 | 10.7 | 4.7 | 3.4 | 6.0 | 4.9 |
| Qwen2.5-VL (72B) | 14.5 | 13.9 | 14.5 | 21.7 | 19.0 | 42.4 | 11.1 | 6.5 | 10.4 | 8.6 | 10.1 | 9.6 | 13.1 | 11.5 | 16.2 |
| GLM 4.5 V (106B) | 15.0 | 13.9 | 16.1 | 17.5 | 20.6 | 23.8 | 8.9 | 7.8 | 10.5 | 13.3 | 13.6 | 25.9 | 13.1 | 13.0 | 18.0 |
| Average | 11.3 | 10.6 | 12.1 | 14.7 | 15.3 | 23.5 | 7.1 | 6.1 | 8.5 | 8.6 | 11.5 | 13.4 | 9.9 | 10.2 | 13.0 |
| *Open-Weight Unified MLLMs (Understanding & Generation)* | | | | | | | | | | | | | | | |
| Bagel (7B) | 9.7 | 5.0 | 11.2 | 7.9 | 0.0 | 7.9 | 3.5 | 5.3 | 4.4 | 12.3 | 4.8 | 13.5 | 7.5 | 4.7 | 8.8 |
| Janus-Pro (7B) | 2.5 | 11.2 | 9.0 | 0.0 | 4.8 | 0.0 | 4.0 | 8.8 | 6.2 | 11.2 | 5.3 | 6.9 | 4.9 | 8.9 | 7.2 |

promising candidates for future autonomous Visual-CoT. We evaluate Bagel (Deng et al., 2025) and Janus-pro (Chen et al., 2025b). Their performance in the simulated Visual-CoT setting provides a proxy for their potential to leverage self-generated visual aids.

For transparency, for all models, we will report the specific version or API endpoint used in Appendix A (*e.g.*, gpt-4o-2024-11-20), a standard practice for reproducibility in benchmark papers. Here, the division of models into "understanding-focused" versus "unified" is not merely descriptive; it constitutes an implicit experimental axis. It is plausible to hypothesize that unified models, even without explicit fine-tuning for such tasks, may demonstrate a greater ability to integrate new visual information in the simulated Visual-CoT setting, as their architectures are designed for tighter coupling between vision and language generation. Thus, by comparing the performance gain between Text-CoT and Visual-CoT for these two classes of open-weight models, we can uncover architectural nuances that favor Visual-CoT-style reasoning.

**Evaluation.** We use micro-averaged accuracy as our metric and employ a tiered pipeline to robustly extract answers from outputs containing lengthy reasoning. Our process prioritizes a fast, rule-based extraction, first attempting to parse a definitive answer from within an <answer></answer>tag, and then falling back to a set of heuristic regular expressions for common answer phrasings. For any remaining ambiguous outputs that elude these deterministic methods, we use a powerful LLM (gpt-4o-2024-11-20) as a semantic judge to analyze the full response and determine the correctness of the final answer against the ground truth. For the LLM evaluation prompts, please refer to Appendix B.

## 4.2 MAIN RESULTS

In this section, we present a comprehensive comparison of various MLLMs on **MIRA** benchmark, with detailed results in Table 1. Our observations are threefold, which are detailed as follows.

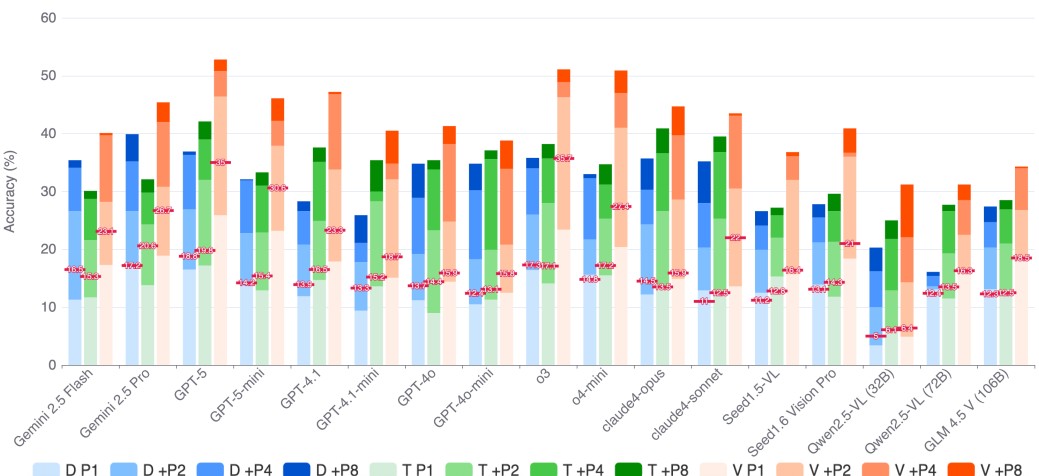

Figure 3: A comprehensive performance comparison of leading models across three evaluation settings: Direct Evaluation (D), Text-CoT Reasoning (T), and Simulated Visual-CoT Reasoning (V). This stacked bar chart shows performance scaling: the base indicates pass@1 accuracy, with segments above capturing gains from pass@2, pass@4, and pass@8. The red horizontal marks show majority voting scores over 8 responses.

**MIRA is challenging, with some categories proving toughest.** Our results show that **MIRA** poses substantial difficulty even for the strongest MLLMs. For example, the latest OpenAI's GPT-5 achieves only average 16.5% accuracy with direct inputs, while there is no a single model achieves accuracy over 20% with only the image and question input. Interestingly, MLLMs generally lags behind on tasks in the Puzzles category, a category that requires meticulous visual understanding and reasoning abilities, compared to other task categories (*i.e.*, average 9.5% on Puzzles *vs.* 16.1% on other categories with the direct input). These observations indicate that the intrinsic challenging nature of **MIRA** and confirm that current MLLMs are adapted for broad general-purpose reasoning but remain poor in handling tasks that demand fine-grained visual comprehension and reasoning.

**Text-CoT alone is not enough for solving MIRA.** Although Text-CoT has proven beneficial on various reasoning benchmarks, our analysis reveals it offers little to no advantage in **MIRA**. In fact, for strong models like Gemini 2.5 Pro and o3, Text-CoT actually degrades performance by a relative percentage of 18.3% and 14.0%, respectively. This trend suggests that stronger models with inherent reasoning capabilities may be adversely impacted by the standard Text-CoT approach. Moreover, on harder categories such as Puzzles and Causal, which evaluate a model's capacity for detailed visual reasoning, Text-CoT harms the accuracy of proprietary models by a relative average proportion of 4.2% and 2.6% severally. These findings not only highlight the limitations of relying exclusively on Text-CoT when problems in **MIRA** demand auxiliary visual processes but also establish the need for a more solid and effective visual reasoning framework.

**The annotated Visual-CoT might be the (temporary) solution.** One solution we proposed for examples that require visual thinking is incorporating human annotated visual demonstrations, which yields substantial improvements across nearly all models. GPT-5-mini, for instance, improves markedly from 13.7% to 23.2%, while all models achieve notable gains with an average relative 33.7% score boost (*e.g.*, 12.2% to 16.3%). Importantly, Visual-CoT particularly benefits challenging categories: Puzzle tasks see a modest increase of 1.0% over the original 9.5% accuracy for private models, whereas Physics tasks experience a striking jump from 20.7% to 40.0% when our visual reasoning patterns are introduced. Open-weight models such as the Qwen2.5-VL family and GLM-4.5V also improve with Visual-CoT (*e.g.*, average 9.9% with direct input *vs.* 13.0% with Visual-CoT), though their gains are more limited. This relatively weak performance is likely due to smaller parameter scales and the lack of extensive training on interleaved Visual-CoT data. Although current unified models like Bagel and Janus-Pro can generate both images and text, they cannot produce images while answering questions. Visual-CoT helps them better understand the question and reason effectively, yielding relative gains of 17.3% and 46.9% for Bagel and Janus-Pro. Overall, while our findings highlight the promise of Visual-CoT as an effective means of enhancing MLLM

Table 2: Comparison of Text-CoT reasoning performance: General Template ($T_{gen}$) vs. Specialized Template ($T_{spec}$). The $\Delta$ (Gain) column indicates the performance improvement when using a specialized template over a general one.

| Model | EG (Geometry) | | | PBR (Physics) | | | ASLP (Puzzles) | | | CT (Causal) | | | Overall | | |
|---|---|---|---|---|---|---|---|---|---|---|---|---|---|---|---|
| | $\mathbf{T}_{gen}$ | $\mathbf{T}_{spec}$ | $\Delta$ | $\mathbf{T}_{gen}$ | $\mathbf{T}_{spec}$ | $\Delta$ | $\mathbf{T}_{gen}$ | $\mathbf{T}_{spec}$ | $\Delta$ | $\mathbf{T}_{gen}$ | $\mathbf{T}_{spec}$ | $\Delta$ | $\mathbf{T}_{gen}$ | $\mathbf{T}_{spec}$ | $\Delta$ |
| *Closed-Source SOTA MLLMs* | | | | | | | | | | | | | | | |
| Claude 4 Opus | 15.6 | 18.0 | +2.4 | 22.2 | 19.0 | -3.2 | 7.8 | 14.5 | +6.7 | 11.6 | 14.2 | +2.6 | 14.3 | 16.4 | +2.1 |
| Claude 4 Sonnet | 10.0 | 16.5 | +6.5 | 18.6 | 15.9 | -2.7 | 11.0 | 11.8 | +0.8 | 15.1 | 11.8 | -3.3 | 13.7 | 14.0 | +0.3 |
| Gemini 2.5 Flash | 11.7 | 11.7 | 0.0 | 22.9 | 28.2 | +5.3 | 5.9 | 9.8 | +3.9 | 12.0 | 14.0 | +2.0 | 13.1 | 15.9 | +2.8 |
| Gemini 2.5 Pro | 11.1 | 12.0 | +0.9 | 27.1 | 28.0 | +0.9 | 7.1 | 8.0 | +0.9 | 17.0 | 17.6 | +0.6 | 15.6 | 16.4 | +0.8 |
| GPT-4.1-mini | 8.9 | 10.6 | +1.7 | 22.2 | 31.4 | +9.2 | 12.5 | 10.2 | -2.3 | 15.4 | 19.8 | +4.4 | 14.8 | 18.0 | +3.2 |
| GPT-4.1 | 17.8 | 16.1 | -1.7 | 16.5 | 17.5 | +1.0 | 7.9 | 15.6 | +7.7 | 17.9 | 13.9 | -4.0 | 15.0 | 15.8 | +0.8 |
| GPT-4o | 11.1 | 11.1 | 0.0 | 11.1 | 6.3 | -4.8 | 3.2 | 6.4 | +3.2 | 12.1 | 12.9 | +0.8 | 9.4 | 9.2 | -0.2 |
| GPT-5 | 14.4 | 15.6 | +1.2 | 22.2 | 33.8 | +11.6 | 15.7 | 13.7 | -2.0 | 19.3 | 19.1 | -0.2 | 17.9 | 20.6 | +2.7 |
| GPT-5-mini | 10.6 | 15.6 | +5.0 | 21.3 | 28.6 | +7.3 | 10.8 | 9.3 | -1.5 | 13.1 | 12.4 | -0.7 | 14.0 | 16.5 | +2.5 |
| o3 | 13.3 | 12.2 | -1.1 | 16.9 | 21.9 | +5.0 | 8.5 | 8.8 | +0.3 | 20.2 | 18.1 | -2.1 | 14.7 | 15.3 | +0.6 |
| o4-mini | 13.1 | 13.3 | +0.2 | 30.5 | 22.6 | -7.9 | 11.4 | 18.8 | +7.4 | 14.4 | 16.3 | +1.9 | 17.4 | 17.8 | +0.4 |
| Seed1.5-VL | 10.6 | 11.7 | +1.1 | 28.6 | 29.4 | +0.8 | 11.2 | 11.8 | +0.6 | 18.0 | 18.5 | +0.5 | 17.1 | 17.9 | +0.8 |
| Seed1.6 Vision Pro | 11.1 | 12.0 | +0.9 | 22.2 | 23.0 | +0.8 | 8.5 | 9.2 | +0.7 | 10.2 | 10.8 | +0.6 | 13.0 | 13.8 | +0.8 |
| Average | 12.2 | 13.6 | +1.4 | 21.0 | 23.6 | +2.6 | 9.2 | 10.8 | +1.6 | 15.2 | 15.3 | +0.1 | 14.4 | 15.8 | +1.4 |
| *Open-Weight MLLMs (Understanding)* | | | | | | | | | | | | | | | |
| Qwen2.5-VL (32B) | 3.9 | 8.0 | +4.1 | 6.4 | 5.6 | -0.8 | 3.9 | 1.9 | -2.0 | 10.7 | 18.6 | +7.9 | 6.0 | 7.8 | +1.8 |
| Qwen2.5-VL (72B) | 13.9 | 11.5 | -2.4 | 19.0 | 16.7 | -2.3 | 6.5 | 8.9 | +2.4 | 10.1 | 13.4 | +3.3 | 11.5 | 11.8 | +0.3 |
| GLM 4.5 V (106B) | 13.9 | 18.1 | +4.2 | 20.6 | 25.0 | +4.4 | 7.8 | 8.3 | +0.5 | 13.6 | 14.4 | +0.8 | 13.0 | 15.3 | +2.3 |
| Average | 10.6 | 12.5 | +2.0 | 15.3 | 15.8 | +0.4 | 6.1 | 6.4 | +0.3 | 11.5 | 15.5 | +4.0 | 10.2 | 11.6 | +1.5 |

performance, they also suggest it is only a temporary solution. Closing this gap will require new training paradigms and datasets that seamlessly integrate visual and textual reasoning.

## 4.3 ATTEMPTS TO PROBE THE MODEL UPPER-BOUND

To evaluate models' "best-case potential" beyond single-answer accuracy to determine whether their failures are due to *accidental reasoning errors* or a *fundamental lack of capability*. We broaden the decoding search space using Pass@$k$ and majority voting (Wang et al., 2022), and further explore model inputs with task-specific prompts aligned to our Visual-CoT.

**Broaden the searching space by Pass@$k$ and majority voting scores.** We employ Pass@$k$ (*e.g.*, $k$=1,2,4,8) to aggregate sampled model answers. where the model generates $k$ different reasoning paths and answers for the same problem, and is considered successful if at least one is correct. We found that while performance increases with $k$ from 1 to 4 with an average 15.3% improvement over all models, the gains nearly converge between $k$=4 and $k$=8 (*i.e.*, only average 3.0%). More concretely, models like Gemini 2.5 Flash and GPT-5 exhibit only marginal gains from pass@4 to pass@8 (*e.g.*, 1.3% and 0.6%, respectively). These results prove that tasks in **MIRA** are highly challenging for these models even with wider search spaces.

We also perform majority voting (Wang et al., 2022) on the eight sampled responses, with the results presented inside red bars in Figure 3. It is noteworthy that, both Pass@$k$ and majority voting see fewer performance gains for models with stronger reasoning abilities — *e.g.*, the stronger GPT-5 shows a 20.4% improvement from Pass@1 to Pass@8, while the slightly less powerful GPT-4o gains more at 23.6%. Similarly, majority voting improves Gemini 2.5 Flash's score by 5.1%, whereas the more advanced Gemini 2.5 Pro only sees a minimal 0.3% increase. While a wider search space benefits weaker models by offering more chances to succeed, stronger models show that their failures are not due to simple reasoning errors but rather a fundamental lack of capability on **MIRA**. This suggests that **MIRA** requires a core reasoning skill that is absent, regardless of how many attempts are made.

**Design Text-CoT to align with Visual-CoT.** While vanilla Text-CoT prompts only instruct models to "think step-by-step", the proposed Visual-CoT provides richer and more explicit reasoning paths for MLLMs to follow. To bridge the gap between Text-CoT and Visual-CoT, we design task-specific CoT prompts that better align with the guidance provided by Visual-CoTs across different tasks (see Appendix B for details). As shown in Table 2, replacing generic prompts with specialized ones leads to consistent improvements, yielding an average gain of 1.4% over closed-source models and 1.5% over three open-weight MLLMs. However, these gains remain modest or even negative with a

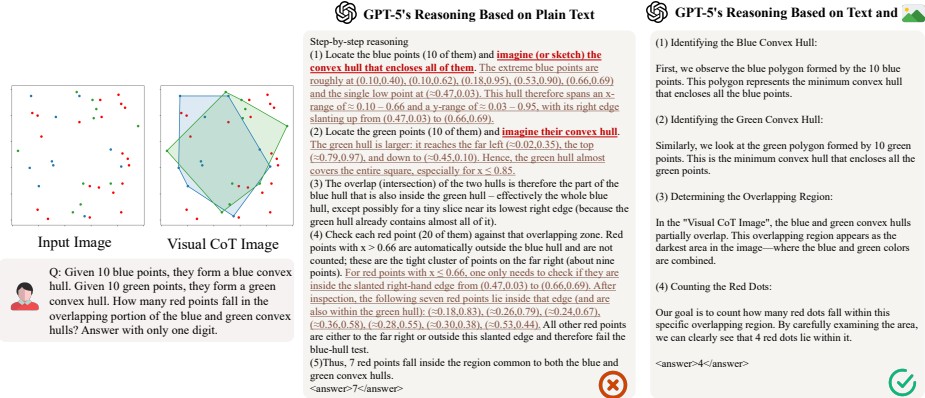

Figure 4: A representative failure case of Text-CoT on a Euclidean Geometry (EG) reasoning task. Even the strongest model (GPT-5) struggles to correctly reason through the problem using plain text, due to its inability to manipulate intermediate visual states. In contrast, the Visual-CoT approach, which leverages intermediate visualizations, enables more accurate localization of the overlapping region and correct counting of red points.

range from -0.2% to +3.2% compared to the stronger improvements brought by Visual-CoT, which achieves an average gain of 4.7% on closed models. This observation underscores the inherent limitation of text-only guidance: certain reasoning steps require visual information that text alone cannot fully capture. These findings further highlight the importance of developing models capable of true visual reasoning and shed light on promising directions for advancing MLLMs.

## 4.4 CASE STUDY

To illustrate the limitations of Text-CoT on **MIRA** tasks, we conduct a case study from the Euclidean Geometry (EG) category (Figure 4). The task asks the model to identify convex hulls of two point sets, locate their overlapping region, and count red dots within it: a process requiring complex spatial reasoning that is nearly impossible to convey in language alone. Even GPT-5 struggles: its text-only reasoning (brown section) attempts to verbally construct and intersect geometric shapes, resulting in vague and unreliable logic. Without visual grounding, it cannot accurately delineate the overlap, often resorting to words like "imagine" (red section), which highlight the inadequacy of purely textual reasoning. In contrast, supplying a simple "Visual-CoT Image" with hand-drawn hulls transforms the task. As shown on the right of Figure 4, the visual cue shifts reasoning from abstract descriptions to direct analysis: GPT-5 clearly identifies the two hulls, marks their intersection, and correctly counts the 4 red dots inside. This case clearly proves that for many visual reasoning tasks, visual grounding is indispensable, and a drawing on a scratchpad can potentially allow the model to reason more accurately.

## 5 CONCLUSION

This paper introduces the **MIRA** benchmark for systematically evaluating the capabilities of MLLMs in complex reasoning scenarios that require the generation of intermediate visual images. The experiments demonstrate three key findings. First, the purely Text-CoT has an intrinsic, medium-level limitation on visually-intensive tasks that is difficult to overcome with prompt engineering, and gains for even state-of-the-art models are very limited. Second, Visual-CoT, which provides intermediate visual clues, yields significant gains across various tasks, with an average relative improvement of over 33%, highlighting the critical role of visual information in complex reasoning. Third, a significant gap remains between closed-source and open-weight models in their ability to effectively utilize visual clues. Overall, MLLMs that rely solely on textual reasoning struggle to address many real-world problems. There is an urgent need for a unified multimodal paradigm geared towards "thinking while drawing" one that generates high-quality intermediate visual states during the reasoning process and tightly couples them with subsequent language-based reasoning, while also pushing for open-weight models to catch up in capability. **MIRA** provides a reproducible evaluation platform and metric system for the development and comparison of such methods.

## ETHICS STATEMENT

All authors have read and comply with the ICLR Code of Ethics. This study does not involve human subjects or sensitive data, and we are unaware of any potential misuse, harm, or bias. No conflicts of interest or compromising sponsorships exist. To mitigate potential risks during dataset curation we inspect every example and make sure that they do not contain personally identifying information or sensitive information that may cause privacy leakage or misuse. Items failing these criteria were excluded. Our intent in releasing **MIRA** is to support research-focused evaluation of model capabilities. According to this intent, any public release will include clear usage terms and guidance that discourage malicious applications (*e.g.*, recommending access only to vetted researchers, providing redacted versions where appropriate, and documenting responsible use).

## REPRODUCIBILITY STATEMENT

We provide detailed descriptions of the benchmark construction, filtering and optimization procedures, as well as the evaluation metrics in Section 3. A comprehensive description of the parameter settings used in model evaluations can be found in Appendix A.

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

# APPENDIX

## A  EXPERIMENTAL MODEL SETTINGS

This section details the configurations for all models evaluated in our experiments. For all API-based models, we utilized the default decoding settings provided by each endpoint, with the maximum output length set to 16,384 tokens. The specific versions and checkpoints are organized in Table 3.

For a specific subset of models, we used tailored generation parameters. Specifically, for Qwen-VL-Max (325B), GLM-4.5V (106B), and both variants of Qwen2.5-VL (32B/72B), we set the maximum output length to 8,192 tokens and used a $\text{top}_p$ value of 1.0. For the Bagel (operating in thinking mode) and Janus-Pro models, we followed the official inference configurations from their respective code repositories to ensure faithful evaluation.

Table 3: A comprehensive list of the models evaluated in our experiments. For all API-based models, the default decoding settings were used, as no specific sampling parameters (e.g., temperature) were set.

| Model | Creator | Version / Checkpoint |
|---|---|---|
| GPT-5 | OpenAI | `gpt-5-2025-08-07` |
| GPT-5-mini | OpenAI | `gpt-5-mini-2025-08-07` |
| GPT-4.1 | OpenAI | `gpt-4.1-2025-04-14` |
| GPT-4.1-mini | OpenAI | `gpt-4.1-mini-2025-04-14` |
| GPT-4o | OpenAI | `gpt-4o-2024-11-20` |
| GPT-4o-mini | OpenAI | `gpt-4o-mini-2024-07-18` |
| o3 | OpenAI | `o3-2025-04-16` |
| o4-mini | OpenAI | `o4-mini-2025-04-16` |
| Claude 4 Opus | Anthropic | `claude4-opus` |
| Claude 4 Sonnet | Anthropic | `claude4-sonnet` |
| Gemini 2.5 Pro | Google | `gemini-2.5-pro` |
| Gemini 2.5 Flash | Google | `gemini-2.5-flash-preview-05-20` |
| Seed1.6 Vision Pro | ByteDance | `doubao-seed-1.6-vision-250815` |
| Seed1.5-VL | ByteDance | `doubao-1.5-vision-pro-250328` |
| Qwen-VL-Max | Alibaba | `qwen-vl-max-0813` |
| Qwen-2.5-VL (32B) | Alibaba | `qwen2.5-vl-32b-instruct` |
| Qwen-2.5-VL (72B) | Alibaba | `qwen2.5-vl-70b-instruct` |
| GLM 4.5V (106B) | ZAI | `glm-4.5v` |

## B  PROMPT SETTINGS

This section provides the specific prompt templates used for the three evaluation levels described in Section 3.2, as well as the specialized templates used for the upgraded Text-CoT analysis in Section 4.3. We also include, at the end, the prompt provided to gpt-4o-2024-11-20 during our evaluation.

**Level 1: Direct Evaluation.** For the direct evaluation setting, a straightforward prompt was used to ask the model for the final answer without requesting intermediate reasoning steps. The template was:

> [Input Image]
> prompt = "'Question: {question}
> Please provide the final answer directly. The final answer is placed in <answer></answer>.'"

**Level 2: Text-CoT Reasoning.** This level tested the models' ability to use text-based reasoning on **MIRA** tasks. The model is prompted to first generate a textual chain of thought and then provide the final answer. Two types of templates were employed to investigate the efficacy of this approach:

- **General Template** ($T_{gen}$): This approach used a generic CoT prompt for all tasks. It served as a baseline to measure the general applicability of text-based reasoning.

> [Input Image]
> prompt = "'Question: {question}
> Please first conduct step-by-step reasoning, and then provide the final answer. The final answer is placed in <answer></answer>.'"

- **Specialized Template** ($T_{spec}$): To test the upper-bound performance of text-only reasoning, dedicated and task-specific CoT prompt templates were designed for each of the 20 tasks in the MIRA dataset. Below are the specific prompts used for each task, organized by category.

(1) EUCLIDEAN GEOMETRY (EG)

**Task: Convex Hull**

> [Input Image]
> prompt = "'This is a convex hull problem. Analyze the points and determine the vertices of the convex hull. Question: {question}
> Please reason step-by-step: 1. Start with one color (e.g., Red): - Visually/algorithmically assess which Red points are extreme (cannot be expressed as a convex combination of other points). - Count how many target points this Red convex hull would contain (on the boundary or strictly inside). Note any collinear runs along edges and whether intermediate collinear points should be kept or skipped per the task convention. 2. Switch to the other color (e.g., Blue): - Repeat the same analysis: identify extreme Blue points and count how many target points the Blue convex hull contains. 3. Cross-check and reconcile: - Compare Red- and Blue-based findings; verify no interior point is mistakenly classified as a hull vertex. - Use supporting checks (orientation tests/cross products) to confirm each candidate vertex lies on the outer boundary; handle collinearity consistently (keep only endpoints unless the problem requires listing all boundary points). 4. Construct the final hull: - Order vertices counterclockwise starting from the leftmost-lowest point (or another clear anchor) and ensure the polygon is simple and closed. - Provide the set/list of hull vertices (by labels or coordinates) and the total count. 5. Briefly justify: - Summarize why each listed vertex is extreme and why excluded points are interior or collinear intermediates.
> The final answer is placed in <answer></answer>.'"

**Task: Overlap**

> [Input Image]
> prompt = "'Choose two images from A–D and overlay them by aligning their black coordinate-axis borders. This produces the overlapping region of the two shapes. Which pair has the largest overlapping area? Output only two letters like 'AC'. Please reason step-by-step: 1. Normalize: confirm all four tiles share the same scale and origin;

treat overlays as perfect border-to-border alignment with no extra rotation/translation. 2. For each pair (AB, AC, AD, BC, BD, CD): - Compare centers and orientations; note how much their silhouettes intersect (heart/square/star/arrow) when placed at identical coordinates. - Use bounding boxes as a quick upper bound; then refine with edge/vertex relationships to judge whether overlap is large (broad interior intersection), medium (partial edge/vertex overlap), or small (mostly disjoint). 3. Track the estimated overlap area (qualitatively or numerically if obvious from symmetry/containment). Resolve ties by preferring the pair with broader interior overlap rather than thin edge contact. 4. State the chosen pair and a 1–2 sentence justification referencing the relative placements/orientations that cause maximal intersection.

The final answer is placed in <answer></answer>.'"

**Task: Localizer**

[Input Image]
prompt = "'Tile the square on the right using the solid-outlined puzzle pieces on the left. Use all pieces; the tiling must be exact—no leftovers, no gaps, no overlaps. Each piece has a circle. After completing the tiling, return the circle coordinates in numerical order using the format: [pieceID, (x, y)]; separate entries with semicolons. Assumptions: use the same unit grid as shown; coordinates are 1-indexed with (x,y) labeled along the top/left axes; rotations and flips are allowed unless forbidden by outlines. Please reason step-by-step: 1. Parse the target grid: record its outer size (width × height) and axes labels. 2. Catalog each piece (1–4): sketch its unit-square footprint, edge types (axis-aligned vs diagonal), and the circle's offset in the piece's local coordinates. 3. Area & boundary check: verify the sum of piece areas equals the target area; note unique constraints (e.g., long diagonals, notches) that can only fit specific borders/corners. 4. Plan placements: anchor the largest/most constrained piece(s) to borders/corners first; ensure diagonals match the grid diagonals; avoid creating unreachable cavities. 5. Place all pieces: finalize positions and orientations so the region is fully covered; confirm no overlaps and all borders align with grid lines/diagonals. 6. Convert circle positions: for each placed piece, transform the circle's local offset to global grid coordinates (x, y) and round to exact grid intersections if applicable. 7. Output strictly in the required order and format: [1, (x1, y1)]; [2, (x2, y2)]; [3, (x3, y3)]; [4, (x4, y4)].

The final answer is placed in <answer></answer>.'"

**Task: Mirror Pattern**

[Input Image]
prompt = "'Which option (A–D) can be obtained by mirroring the original image once? You may follow these steps to reason: 1) Horizontally mirror the original image. 2) After the reflection, allow an arbitrary in-plane rotation and critically compare against each option A–D (match landmark positions/orientations; rule out any option that would require a second reflection or non-rigid warping).

The final answer is placed in <answer></answer>.'"

**Task: Cubes Count**

[Input Image]
prompt = "'What's the number of cubes presented in the image? Please follow these steps: 1. Identify each layer from bottom to top. 2. For each layer, count how many cubes are present. 3. Add up the counts to get the total number of cubes.

The final answer is placed in <answer></answer>.'"

**Task: Cubes Missing**

[Input Image]
prompt = "'What is the number of cubes needed to fill in the structure so that it becomes a solid block with no internal gaps? Please follow these steps: 1. Identify the full dimensions of the solid block (length × width × height). 2. For each layer (from top to bottom), count: - Maximum possible cubes in that layer if solid - Actual cubes present - Missing cubes = (full layer) – (present layer) 3. Add the missing cubes across all layers. The final answer is placed in <answer></answer>.'"

(2) PHYSICS-BASED REASONING (PBR)

**Task: Billiards**

[Input Image]
prompt = "'In the image, a billiards table has pockets labeled 1–6. The blue ball rolls along the green arrow, with no spin, perfectly elastic cushion bounces, and unlimited momentum. Which numbered pocket will it finally enter? Answer with a single digit 1–6. You may follow these steps to reason: 1) Normalize the table: record the ball's starting point and the arrow's direction; pockets are fixed at labels 1–6. 2) Use the mirror (unfolding) method: virtually reflect the table across a cushion each time the path would bounce. Extend the initial ray straight through these mirrored copies until it hits the center of a mirrored pocket. 3) Map that hit back to the original table to identify the real pocket number; equivalently, enforce equal-angles for each bounce and verify the same destination. 4) Output only the pocket label (1–6).
The final answer is placed in <answer></answer>.'"

**Task: Electric Charge**

[Input Image]
prompt = "'Question: {question}

You may follow these steps to reason:

1. Parse the setup: list each charge with sign, magnitude, and coordinates; identify the target object (which charge/point the net force is asked about).

2. For each source charge, determine the force direction on the target (attraction if opposite sign, repulsion if same sign); sketch/describe the vector qualitatively.

3. Compute each force's magnitude with Coulomb's law

$$|F_i| = k \frac{|q_i q_t|}{r_i^2},$$

and compute vector components using the displacement unit vector from source to target.

4. Apply superposition: sum the components $F_x$ and $F_y$ to obtain the net force vector; use symmetry to simplify whenever possible.

5. Report the net magnitude

$$\sqrt{F_x^2 + F_y^2}$$

and direction (angle or cardinal description), and check limiting/special cases (e.g., $r = 0$ excluded, equal/opposite charges cancel along symmetry axes).

The final answer is placed in <answer></answer>.'"

**Task: Mirror Clock**

[Input Image]
prompt = "'Question: {question}

You may reason as follows:

1. First mirror the clock face (by default, a left–right reflection about the vertical axis).

2. Record the hands' angles relative to 12 o'clock after mirroring. The angles transform as
$$\theta' = 360° - \theta,$$
i.e. clockwise and counterclockwise directions swap.

3. If required to match choices/diagram, you may then apply an in-plane rotation (0°, 90°, 180°, 270°), but do *not* perform a second reflection.

4. Convert the mirrored angles back to time and handle hour-minute carry. For minutes $m$ and hours $h$ (12-hour clock, with $h \in \{1, \ldots, 12\}$):
$$m' \equiv (60 - m) \pmod{60},$$
and define the borrow/carry as
$$\text{carry} = \begin{cases} 1, & m \neq 0, \\ 0, & m = 0. \end{cases}$$
Then compute the mirrored hour
$$h' \equiv (12 - h - \text{carry}) \pmod{12}.$$
When presenting the result convert hour 0 to 12 for human-readable 12-hour time.

5. Compare with the choices, state the final time/option, and explain the key correspondences in 1–2 sentences.

The final answer is placed in <answer></answer>.'"

(3) ABSTRACT SPATIAL & LOGICAL PUZZLES (ASLP)

**Task: Unfolded Cube**

[Input Image]
prompt = "'This is a cube unfolding problem. Determine which of the options can be folded into the given cube, or what the unfolded pattern looks like. Explain your spatial reasoning.
Question: {question}
The final answer is placed in <answer></answer>.'"

**Task: Defuse A Bomb**

[Input Image]
prompt = "'Question: {question} You can first connect the lines to the obstructed area and then go through each option one by one to determine which wire to cut.
The final answer is placed in <answer></answer>.'"

**Task: Multi-piece Puzzle**

[Input Image]
prompt = "'Question: {question} You can carefully consider the details of each option before making your choice.
The final answer is placed in <answer></answer>.'"

**Task: Puzzle**

[Input Image]
prompt = "'Given the object above. There is a missing piece in the white area. Which of the five pieces (A, B, C, D, or E) fits perfectly into the missing part of the object? Please examine the immediate surroundings first and work step-by-step: 1. Describe the boundary shape (angles, curves) of the hole. 2. Describe any pattern/stripe/texture crossing the boundary. 3. Note lighting/shading and relative scale. 4. Compare each candidate to steps 1–3 and rule out mismatches. 5. State final choice and a brief justification (3–5 short sentences).
The final answer is placed in <answer></answer>.'"

**Task: Trailer Cubes Count**

[Input Image]
prompt = "'Based on the three views, what's the maximum number of cubes that could be present? Steps: 1. For each column (grid position in the top view), determine the maximum possible height consistent with front and side views. 2. Count cubes in each column = column height. 3. Sum across all columns for the total.
The final answer is placed in <answer></answer>.'"

**Task: Trailer Cubes Missing**

[Input Image]
prompt = "'Given the three views, what is the minimum number of cubes needed to fill in the structure so that it becomes a solid block with no internal gaps?
Procedure the model must use:

1. Read the top view to list allowed (r,c) column positions.

2. Let $H_{full}$ be the required cuboid height (the maximum height implied by front/side).

3. To produce a minimal current 3D consistent with views.

4. For each allowed (r,c) column, compute missing $= H_{full} -$ assigned_height.

5. Sum missing cube values.

The final answer is placed in <answer></answer>.'"

(4) CAUSAL TRANSFORMATIONS (CT)

**Task: Paper Airplane**

[Input Image]
prompt = "'Question: {question}
Please note the differences between the folding positions of the wings, center, and nose of the aircraft in each option, and then choose the appropriate option.
The final answer is placed in <answer></answer>.'"

**Task: Gear Rotation**

[Input Image]
prompt = "'Question: {question}
You can answer this question based on the fact that two connected gears rotate in opposite directions, a conveyor belt rotates in the same direction as the gears, and a crossed conveyor belt rotates in the opposite direction as the gears.
The final answer is placed in <answer></answer>.'"

**Task: Rolling Dice (Top)**

[Input Image]
prompt = "'Question: {question}
You can list the situation of each side of the dice after each roll, mark the top and bottom, and then after you have reasoned through each step, combine each step with the final result and choose the correct option.
The final answer is placed in <answer></answer>.'"

**Task: Rolling Dice (Two)**

[Input Image]
prompt = "'Question: {question}
You can list the situation of each side of the dice after each roll, mark the top and bottom, and then after you have reasoned through each step, combine each step with the final result and choose the correct option.
The final answer is placed in <answer></answer>.'"

**Task: Rolling Dice (Sum)**

[Input Image]
prompt = "'Question: {question}
You can list the situation of each side of the dice after each roll, mark the top and bottom, and then after you have reasoned through each step, combine each step with the final result and choose the correct option.
The final answer is placed in <answer></answer>.'"

**Level 3: Simulated Visual-CoT Reasoning.** This level evaluates the model's ability to utilize visual information in its reasoning process. Given that current MLLMs are unable to generate their own intermediate visual steps, this setting simulates a Visual-CoT process. The model is provided with the initial problem image along with a sequence of manually annotated intermediate images that act as visual clues. The prompt then directs the model to reason based on this sequence of visuals to arrive at the final answer. This approach is designed to measure the performance improvement gained from visual aids and to understand the potential of a true "think while drawing" capability.

[Input Image] [CoT Image 1] [CoT Image 2] ...
prompt = "'Based on the question image and the intermediate reasoning image(s) provided, please continue the reasoning to solve the problem.
Question: {question}
The final answer is placed in <answer></answer>.'"

**Evaluation Prompt.** This prompt is used by an evaluator model to judge the correctness of the primary model's response.

[Input Image]
Judge prompt = "'You are a strict and precise evaluator. Your task is to determine whether the model's final answer is correct based on the ground truth.
Your evaluation must focus exclusively on the answer contained within the <answer></answer>tags, as well as the final answer portion at the end of the model's response. Ignore all reasoning, explanations, or any other text outside of these sections. The correctness of the reasoning process is not part of your evaluation.
Here is the data:
Question: "{question}"
Ground Truth Answer: "{ground truth}"
Model's Full Response: "{model response}"
Based on the ground truth, is the answer inside the <answer>tag correct?

Please respond with only one word: "Correct" or "Incorrect". "'

## C   DETAILED EXPERIMENTAL TABLES

This section provides a detailed breakdown of model performance across all sub-categories within the **MIRA** benchmark, supplementing the main results presented in Table 1. The following tables correspond to Tables 4-10 as referenced in the main paper.

Table 4: Detailed Results for Euclidean Geometry (Convex Hull, Mirror Pattern) and Physics-Based Reasoning (Mirror Clock) Tasks.

| Model | Convex Hull | | | Mirror Pattern | | | Mirror Clock | | |
|---|---|---|---|---|---|---|---|---|---|
| | **D** | **T** | **V** | **D** | **T** | **V** | **D** | **T** | **V** |
| GPT-5 | 16.7 | 16.7 | 20.0 | 26.7 | 33.3 | 26.7 | 23.3 | 33.3 | 46.7 |
| GPT-5-mini | 10.0 | 16.7 | 23.3 | 30.0 | 10.0 | 16.7 | 3.33 | 6.67 | 43.3 |
| GPT-4.1 | 13.3 | 16.7 | 10.0 | 20.0 | 36.7 | 26.7 | 3.33 | 6.67 | 13.3 |
| GPT-4.1-mini | 3.33 | 13.3 | 20.0 | 23.3 | 20.0 | 30.0 | 0.00 | 16.7 | 13.3 |
| GPT-4o | 6.67 | 3.33 | 16.7 | 36.7 | 23.3 | 20.0 | 0.00 | 0.00 | 0.00 |
| GPT-4o-mini | 6.67 | 10.0 | 16.7 | 13.3 | 30.0 | 20.0 | 0.00 | 3.33 | 0.00 |
| o4-mini | 13.8 | 11.5 | 3.33 | 33.3 | 20.0 | 16.7 | 16.7 | 13.3 | 33.3 |
| o3 | 17.9 | 0.00 | 16.7 | 26.7 | 26.7 | 23.3 | 10.0 | 6.67 | 33.3 |
| Claude 4 Opus | 6.67 | 13.3 | 13.3 | 30.0 | 36.7 | 30.0 | 0.00 | 0.00 | 0.00 |
| Claude 4 Sonnet | 16.7 | 10.0 | 13.3 | 26.7 | 23.3 | 20.0 | 6.67 | 3.33 | 6.67 |
| Seed1.5-VL | 13.3 | 13.3 | 16.7 | 16.7 | 16.7 | 26.7 | 0.00 | 0.00 | 16.7 |
| Seed1.6 Vision Pro | 10.0 | 3.33 | 40.0 | 26.7 | 30.0 | 26.7 | 0.00 | 0.00 | 16.7 |
| Gemini 2.5 Flash | 0.00 | 0.00 | 3.33 | 13.3 | 30.0 | 30.0 | 6.67 | 6.67 | 40.0 |
| Gemini 2.5 Pro | 6.67 | 10.0 | 10.0 | 20.0 | 30.0 | 16.7 | 23.3 | 10.0 | 50.0 |
| Qwen-VL-max-latest (325B) | 3.33 | 0.00 | 23.3 | 13.3 | 33.3 | 33.3 | 6.67 | 0.00 | 0.00 |
| Qwen2.5-VL (32B) | 0.00 | 0.00 | 10.0 | 13.3 | 3.33 | 13.3 | 0.00 | 0.00 | 3.33 |
| Qwen2.5-VL (72B) | 16.7 | 20.0 | 16.7 | 20.0 | 20.0 | 30.0 | 3.33 | 0.00 | 3.33 |
| GLM 4.5 V (106B) | 16.7 | 13.3 | 20.0 | 30.0 | 33.3 | 26.7 | 0.00 | 0.00 | 0.00 |
| Bagel (7B) | 0.00 | 0.00 | 10.0 | 30.0 | 16.7 | 30.0 | 0.00 | 0.00 | 0.00 |
| Janus-pro (7B) | 0.00 | 16.7 | 0.00 | 3.33 | 13.3 | 23.3 | 0.00 | 0.00 | 0.00 |

Table 5: Detailed Results for Euclidean Geometry (Overlap), Abstract Puzzles (Unfolded Cube), and Physics-Based Reasoning (Billiards) Tasks.

| Model | Overlap | | | Unfolded Cube | | | Billiards | | |
|---|---|---|---|---|---|---|---|---|---|
| | **D** | **T** | **V** | **D** | **T** | **V** | **D** | **T** | **V** |
| GPT-5 | 36.7 | 36.7 | 46.7 | 8.70 | 27.3 | 45.8 | 23.8 | 9.50 | 85.7 |
| GPT-5-mini | 20.0 | 36.7 | 80.0 | 0.00 | 13.6 | 50.0 | 9.52 | 9.52 | 61.9 |
| GPT-4.1 | 56.7 | 50.0 | 63.3 | 0.00 | 0.00 | 3.85 | 9.52 | 14.3 | 76.2 |
| GPT-4.1-mini | 3.33 | 13.3 | 20.0 | 23.3 | 20.0 | 30.0 | 0.00 | 16.7 | 13.3 |
| GPT-4o | 53.3 | 36.7 | 30.0 | 0.00 | 0.00 | 0.00 | 4.76 | 14.3 | 57.1 |
| GPT-4o-mini | 13.3 | 40.0 | 16.7 | 0.00 | 0.00 | 0.00 | 19.1 | 9.52 | 38.1 |
| o4-mini | 33.3 | 43.3 | 56.7 | 14.3 | 0.00 | 23.1 | 15.8 | 21.1 | 70.0 |
| o3 | 36.7 | 43.3 | 66.7 | 10.0 | 0.00 | 23.5 | 9.52 | 25.0 | 90.5 |
| Claude 4 Opus | 36.7 | 43.3 | 43.3 | 0.00 | 0.00 | 7.69 | 9.52 | 23.8 | 42.9 |
| Claude 4 Sonnet | 23.3 | 26.7 | 53.3 | 0.00 | 0.00 | 0.00 | 9.52 | 9.52 | 42.9 |
| Seed1.5-VL | 36.7 | 33.3 | 46.7 | 0.00 | 7.69 | 3.85 | 9.52 | 23.8 | 52.4 |
| Seed1.6 Vision Pro | 43.3 | 33.3 | 56.7 | 0.00 | 7.69 | 7.69 | 19.1 | 19.1 | 85.7 |
| Gemini 2.5 Flash | 36.7 | 30.0 | 46.7 | 0.00 | 3.85 | 0.00 | 9.52 | 14.3 | 52.4 |
| Gemini 2.5 Pro | 36.7 | 20.0 | 63.3 | 19.2 | 3.85 | 7.69 | 28.6 | 14.3 | 61.9 |
| Qwen-VL-max-latest (325B) | 46.7 | 43.3 | 46.7 | 6.67 | 0.00 | 3.33 | 14.3 | 19.1 | 57.1 |
| Qwen2.5-VL (32B) | 13.3 | 20.0 | 10.0 | 0.00 | 0.00 | 0.00 | 14.3 | 19.1 | 9.52 |
| Qwen2.5-VL (72B) | 40.0 | 43.3 | 40.0 | 0.00 | 0.00 | 0.00 | 4.76 | 9.52 | 76.2 |
| GLM 4.5 V (106B) | 40.0 | 33.3 | 46.7 | 0.00 | 3.85 | 0.00 | 9.52 | 9.52 | 38.1 |
| Bagel (7B) | 10.0 | 13.3 | 16.7 | 0.00 | 0.00 | 0.00 | 19.1 | 0.00 | 19.1 |
| Janus-pro (7B) | 0.00 | 20.0 | 20.0 | 0.00 | 0.00 | 0.00 | 4.76 | 4.76 | 0.00 |

Table 6: Detailed Results for Euclidean Geometry (Localizer), Causal Transformations (Paper Airplane), and Abstract Puzzles (Defuse A Bomb) Tasks.

| Model | Localizer | | | Paper Airplane | | | Defuse A Bomb | | |
|---|---|---|---|---|---|---|---|---|---|
| | D | T | V | D | T | V | D | T | V |
| GPT-5 | 0.00 | 0.00 | 0.00 | 32.0 | 32.0 | 28.0 | 32.0 | 28.0 | 32.0 |
| GPT-5-mini | 0.00 | 0.00 | 0.00 | 12.0 | 16.0 | 36.0 | 16.0 | 28.0 | 16.0 |
| GPT-4.1 | 0.00 | 0.00 | 0.00 | 28.0 | 16.0 | 16.0 | 24.0 | 28.0 | 40.0 |
| GPT-4.1-mini | 0.00 | 0.00 | 0.00 | 24.0 | 20.0 | 28.0 | 28.0 | 32.0 | 20.0 |
| GPT-4o | 0.00 | 0.00 | 0.00 | 20.0 | 20.0 | 24.0 | 16.0 | 8.00 | 24.0 |
| GPT-4o-mini | 0.00 | 0.00 | 0.00 | 20.0 | 12.0 | 20.0 | 8.00 | 24.0 | 32.0 |
| o4-mini | 0.00 | 0.00 | 0.00 | 20.0 | 12.0 | 40.0 | 24.0 | 24.0 | 28.0 |
| o3 | 0.00 | 0.00 | 0.00 | 28.0 | 32.0 | 28.0 | 28.0 | 24.0 | 12.0 |
| Claude 4 Opus | 0.00 | 0.00 | 0.00 | 28.0 | 20.0 | 12.0 | 20.0 | 24.0 | 32.0 |
| Claude 4 Sonnet | 0.00 | 0.00 | 0.00 | 16.0 | 12.0 | 12.0 | 28.0 | 36.0 | 28.0 |
| Seed1.5-VL | 0.00 | 0.00 | 0.00 | 4.00 | 24.0 | 16.0 | 32.0 | 40.0 | 8.00 |
| Seed1.6 Vision Pro | 0.00 | 0.00 | 0.00 | 20.0 | 24.0 | 24.0 | 28.0 | 20.0 | 8.00 |
| Gemini 2.5 Flash | 0.00 | 0.00 | 0.00 | 16.0 | 12.0 | 24.0 | 16.0 | 16.0 | 12.0 |
| Gemini 2.5 Pro | 0.00 | 0.00 | 0.00 | 28.0 | 32.0 | 20.0 | 20.0 | 16.7 | 28.0 |
| Qwen-VL-max-latest (325B) | 0.00 | 0.00 | 0.00 | 20.0 | 4.00 | 20.0 | 32.0 | 16.0 | 36.0 |
| Qwen2.5-VL (32B) | 0.00 | 0.00 | 0.00 | 12.0 | 12.0 | 16.0 | 0.00 | 0.00 | 4.00 |
| Qwen2.5-VL (72B) | 0.00 | 0.00 | 0.00 | 20.0 | 4.00 | 20.0 | 32.0 | 16.0 | 36.0 |
| GLM 4.5 V (106B) | 0.00 | 0.00 | 0.00 | 4.00 | 12.0 | 12.0 | 30.4 | 20.0 | 28.0 |
| Bagel (7B) | 0.00 | 0.00 | 0.00 | 28.0 | 12.0 | 12.0 | 16.0 | 0.00 | 20.0 |
| Janus-pro (7B) | 0.00 | 0.00 | 0.00 | 4.00 | 8.00 | 12.0 | 16.0 | 0.00 | 8.00 |

Table 7: Detailed Results for Abstract Puzzles (Multi-piece Puzzle), Physics-Based Reasoning (Electric Charge), and Causal Transformations (Rolling Dice: Top) Tasks.

| Model | Multi-piece Puzzle | | | Electric Charge | | | Rolling Dice: Top | | |
|---|---|---|---|---|---|---|---|---|---|
| | D | T | V | D | T | V | D | T | V |
| GPT-5 | 0.00 | 0.00 | 6.67 | 42.7 | 23.8 | 28.6 | 30.8 | 30.8 | 92.3 |
| GPT-5-mini | 0.00 | 0.00 | 0.00 | 71.4 | 47.6 | 14.3 | 38.7 | 26.9 | 73.1 |
| GPT-4.1 | 0.00 | 0.00 | 0.00 | 23.8 | 28.6 | 28.6 | 11.5 | 26.9 | 26.9 |
| GPT-4.1-mini | 0.00 | 3.33 | 0.00 | 28.6 | 33.3 | 66.7 | 3.85 | 23.1 | 23.1 |
| GPT-4o | 0.00 | 0.00 | 3.33 | 19.1 | 19.1 | 57.1 | 15.4 | 7.69 | 11.5 |
| GPT-4o-mini | 0.00 | 0.00 | 0.00 | 23.8 | 4.76 | 14.3 | 19.2 | 26.9 | 30.8 |
| o4-mini | 6.90 | 3.57 | 0.00 | 23.8 | 57.1 | 28.6 | 21.1 | 15.0 | 57.7 |
| o3 | 0.00 | 0.00 | 3.57 | 47.6 | 19.1 | 19.1 | 30.8 | 26.9 | 76.9 |
| Claude 4 Opus | 3.33 | 0.00 | 0.00 | 47.6 | 42.9 | 42.9 | 11.5 | 15.4 | 23.1 |
| Claude 4 Sonnet | 3.33 | 3.33 | 0.00 | 42.9 | 42.9 | 33.3 | 15.4 | 34.6 | 30.8 |
| Seed1.5-VL | 0.00 | 0.00 | 0.00 | 52.4 | 61.9 | 61.9 | 23.1 | 38.5 | 19.2 |
| Seed1.6 Vision Pro | 0.00 | 3.33 | 0.00 | 42.9 | 47.6 | 52.4 | 15.4 | 26.9 | 26.9 |
| Gemini 2.5 Flash | 0.00 | 0.00 | 3.33 | 42.9 | 47.6 | 47.6 | 11.5 | 19.2 | 11.5 |
| Gemini 2.5 Pro | 3.33 | 6.67 | 3.33 | 71.4 | 57.1 | 66.7 | 7.69 | 23.1 | 15.4 |
| Qwen-VL-max-latest (325B) | 3.33 | 0.00 | 0.00 | 52.4 | 47.6 | 38.1 | 19.2 | 19.2 | 38.5 |
| Qwen2.5-VL (32B) | 0.00 | 0.00 | 0.00 | 0.00 | 0.00 | 0.00 | 7.69 | 11.5 | 7.69 |
| Qwen2.5-VL (72B) | 0.00 | 0.00 | 3.33 | 57.1 | 47.6 | 47.6 | 23.1 | 11.5 | 23.1 |
| GLM 4.5 V (106B) | 0.00 | 3.33 | 0.00 | 42.9 | 52.4 | 33.3 | 19.2 | 26.9 | 61.5 |
| Bagel (7B) | 0.00 | 0.00 | 0.00 | 4.76 | 0.00 | 0.00 | 15.4 | 3.85 | 19.2 |
| Janus-pro (7B) | 0.00 | 0.00 | 0.00 | 0.00 | 14.3 | 0.00 | 11.5 | 3.85 | 11.5 |

# D   DATASET SHOWCASE

The **MIRA** benchmark is composed of 546 multimodal problems spanning 20 distinct task types. These tasks are curated to be challenging and require intermediate visual reasoning, a process analogous to how humans "draw to think" to solve complex problems. The tasks fall into four challenging domains: Euclidean Geometry (EG), Physics-Based Reasoning (PBR), Abstract Spatial & Logical Puzzles (ASLP), and Causal Transformations (CT).

To supplement the overview provided in Figure 1 and offer a more intuitive understanding of the dataset, we showcase several representative examples for each category below (Figure 5- 14).

Table 8: Detailed Results for Causal Transformations Tasks (Rolling Dice: Sum, Rolling Dice: Two, Gear Rotation).

| Model | Rolling Dice: Sum | | | Rolling Dice: Two | | | Gear Rotation | | |
|---|---|---|---|---|---|---|---|---|---|
| | D | T | V | D | T | V | D | T | V |
| GPT-5 | 11.5 | 3.85 | 7.96 | 0.00 | 0.00 | 0.00 | 15.0 | 30.0 | 15.0 |
| GPT-5-mini | 15.4 | 7.69 | 3.85 | 0.00 | 0.00 | 0.00 | 20.0 | 15.0 | 10.0 |
| GPT-4.1 | 11.5 | 11.5 | 3.85 | 0.00 | 0.00 | 0.00 | 15.0 | 35.0 | 30.0 |
| GPT-4.1-mini | 3.85 | 3.85 | 7.69 | 0.00 | 0.00 | 0.00 | 20.0 | 30.0 | 15.0 |
| GPT-4o | 0.00 | 7.69 | 0.00 | 0.00 | 0.00 | 0.00 | 35.0 | 25.0 | 10.0 |
| GPT-4o-mini | 3.85 | 7.69 | 0.00 | 0.00 | 0.00 | 0.00 | 35.0 | 40.0 | 25.0 |
| o4-mini | 11.8 | 5.00 | 4.17 | 0.00 | 0.00 | 0.00 | 30.0 | 40.0 | 20.0 |
| o3 | 11.5 | 12.0 | 7.69 | 0.00 | 0.00 | 0.00 | 30.0 | 30.0 | 25.0 |
| Claude 4 Opus | 3.85 | 7.69 | 15.4 | 0.00 | 0.00 | 0.00 | 20.0 | 15.0 | 10.0 |
| Claude 4 Sonnet | 11.5 | 3.85 | 0.00 | 0.00 | 0.00 | 0.00 | 20.0 | 25.0 | 5.00 |
| Seed1.5-VL | 7.69 | 7.69 | 7.69 | 0.00 | 0.00 | 0.00 | 35.0 | 20.0 | 20.0 |
| Seed1.6 Vision Pro | 3.85 | 0.00 | 0.00 | 0.00 | 0.00 | 0.00 | 45.0 | 0.00 | 5.00 |
| Gemini 2.5 Flash | 7.69 | 3.85 | 0.00 | 0.00 | 0.00 | 0.00 | 35.0 | 25.0 | 35.0 |
| Gemini 2.5 Pro | 15.4 | 0.00 | 0.00 | 0.00 | 0.00 | 0.00 | 35.0 | 30.0 | 15.0 |
| Qwen-VL-max-latest (325B) | 0.00 | 0.00 | 0.00 | 0.00 | 0.00 | 7.69 | 30.0 | 15.0 | 35.0 |
| Qwen2.5-VL (32B) | 0.00 | 0.00 | 0.00 | 0.00 | 0.00 | 0.00 | 0.00 | 30.0 | 0.00 |
| Qwen2.5-VL (72B) | 0.00 | 0.00 | 0.00 | 0.00 | 0.00 | 0.00 | 0.00 | 35.0 | 5.00 |
| GLM 4.5 V (106B) | 15.4 | 3.85 | 11.5 | 7.69 | 0.00 | 34.6 | 20.0 | 25.0 | 10.0 |
| Bagel (7B) | 3.85 | 0.00 | 0.00 | 3.33 | 3.33 | 3.33 | 15.4 | 23.1 | 11.5 |
| Janus-pro (7B) | 3.85 | 7.69 | 0.00 | 0.00 | 0.00 | 0.00 | 25.0 | 10.0 | 5.00 |

Table 9: Detailed Results for Euclidean Geometry (Cubes Count, Cubes Missing) and Abstract Puzzles (Puzzle) Tasks.

| Model | Cubes Count | | | Cubes Missing | | | Puzzle | | |
|---|---|---|---|---|---|---|---|---|---|
| | D | T | V | D | T | V | D | T | V |
| GPT-5 | 3.33 | 0.00 | 0.00 | 3.33 | 0.00 | 0.00 | 3.85 | 34.6 | 30.8 |
| GPT-5-mini | 0.00 | 0.00 | 0.00 | 0.00 | 0.00 | 0.00 | 19.2 | 19.2 | 23.1 |
| GPT-4.1 | 0.00 | 0.00 | 0.00 | 6.67 | 3.33 | 0.00 | 11.5 | 15.4 | 15.4 |
| GPT-4.1-mini | 0.00 | 0.00 | 0.00 | 3.33 | 6.67 | 0.00 | 23.1 | 15.4 | 15.4 |
| GPT-4o | 3.33 | 0.00 | 3.33 | 3.33 | 3.33 | 0.00 | 11.5 | 11.5 | 30.8 |
| GPT-4o-mini | 3.33 | 0.00 | 6.67 | 6.67 | 3.33 | 6.67 | 34.6 | 15.4 | 23.1 |
| o4-mini | 3.57 | 4.00 | 7.14 | 0.00 | 0.00 | 0.00 | 30.8 | 23.1 | 19.2 |
| o3 | 3.33 | 6.67 | 0.00 | 6.67 | 3.33 | 3.33 | 26.9 | 23.1 | 34.6 |
| Claude 4 Opus | 0.00 | 0.00 | 0.00 | 3.33 | 0.00 | 3.33 | 19.2 | 23.1 | 23.1 |
| Claude 4 Sonnet | 3.33 | 0.00 | 3.33 | 3.33 | 0.00 | 0.00 | 30.8 | 26.9 | 23.1 |
| Seed1.5-VL | 0.00 | 0.00 | 3.33 | 0.00 | 0.00 | 3.33 | 3.85 | 7.69 | 7.69 |
| Seed1.6 Vision Pro | 0.00 | 0.00 | 3.33 | 0.00 | 0.00 | 3.33 | 11.5 | 0.00 | 3.85 |
| Gemini 2.5 Flash | 3.33 | 0.00 | 0.00 | 3.33 | 10.0 | 13.3 | 19.2 | 15.4 | 19.2 |
| Gemini 2.5 Pro | 0.00 | 6.67 | 0.00 | 0.00 | 0.00 | 0.00 | 11.5 | 11.5 | 19.2 |
| Qwen-VL-max-latest (325B) | 6.67 | 0.00 | 3.33 | 0.00 | 0.00 | 0.00 | 30.8 | 34.6 | 23.1 |
| Qwen2.5-VL (32B) | 0.00 | 0.00 | 0.00 | 0.00 | 0.00 | 0.00 | 7.69 | 23.1 | 19.2 |
| Qwen2.5-VL (72B) | 6.67 | 0.00 | 0.00 | 3.33 | 0.00 | 0.00 | 30.8 | 23.1 | 23.1 |
| GLM 4.5 V (106B) | 0.00 | 0.00 | 0.00 | 3.33 | 3.33 | 3.33 | 19.2 | 15.4 | 23.1 |
| Bagel (7B) | 0.00 | 0.00 | 3.33 | 3.33 | 3.33 | 3.33 | 15.4 | 23.1 | 11.5 |
| Janus-pro (7B) | 0.00 | 0.00 | 0.00 | 10.0 | 0.00 | 6.67 | 0.00 | 34.6 | 26.9 |

# E   DISCLOSURE OF LARGE LANGUAGE MODEL USE

As required by ICLR 2026 policy, we report that a large language model (ChatGPT) was used for language refinement of this paper, including improvements to grammar, phrasing, and style.

All research concepts, methods, analyses, and conclusions were conceived and executed solely by the authors. The model's role was confined to copy-editing and it made no contribution to the scientific content. The authors are fully responsible for the final manuscript.

Table 10: Detailed Results for Abstract Puzzles Tasks (Trailer Cubes Count, Trailer Cubes Missing).

| Model | Trailer Cubes Count | | | Trailer Cubes Missing | | |
|---|---|---|---|---|---|---|
| | D | T | V | D | T | V |
| GPT-5 | 16.0 | 4.00 | 4.00 | 4.00 | 0.00 | 0.00 |
| GPT-5-mini | 4.00 | 4.00 | 8.00 | 4.00 | 0.00 | 4.00 |
| GPT-4.1 | 0.00 | 0.00 | 4.00 | 4.00 | 4.00 | 0.00 |
| GPT-4.1-mini | 0.00 | 0.00 | 0.00 | 0.00 | 4.00 | 0.00 |
| GPT-4o | 0.00 | 0.00 | 0.00 | 0.00 | 0.00 | 0.00 |
| GPT-4o-mini | 0.00 | 0.00 | 0.00 | 4.00 | 0.00 | 0.00 |
| o4-mini | 11.8 | 17.7 | 0.00 | 0.00 | 0.00 | 0.00 |
| o3 | 4.00 | 0.00 | 0.00 | 0.00 | 4.00 | 4.00 |
| Claude 4 Opus | 0.00 | 0.00 | 0.00 | 4.00 | 0.00 | 0.00 |
| Claude 4 Sonnet | 0.00 | 0.00 | 0.00 | 0.00 | 0.00 | 0.00 |
| Seed1.5-VL | 16.0 | 12.0 | 0.00 | 0.00 | 0.00 | 4.00 |
| Seed1.6 Vision Pro | 4.00 | 12.0 | 8.00 | 8.00 | 8.00 | 0.00 |
| Gemini 2.5 Flash | 0.00 | 0.00 | 4.00 | 4.00 | 0.00 | 4.00 |
| Gemini 2.5 Pro | 8.00 | 0.00 | 0.00 | 4.00 | 4.00 | 0.00 |
| Qwen-VL-max-latest (325B) | 4.00 | 0.00 | 4.00 | 4.00 | 4.00 | 4.00 |
| Qwen2.5-VL (32B) | 0.00 | 0.00 | 0.00 | 0.00 | 0.00 | 4.00 |
| Qwen2.5-VL (72B) | 0.00 | 0.00 | 0.00 | 4.00 | 0.00 | 0.00 |
| GLM 4.5 V (106B) | 4.00 | 4.00 | 4.00 | 0.00 | 0.00 | 8.00 |
| Bagel (7B) | 0.00 | 16.0 | 0.00 | 0.00 | 4.00 | 4.00 |
| Janus-pro (7B) | 0.00 | 0.00 | 0.00 | 4.00 | 0.00 | 4.00 |

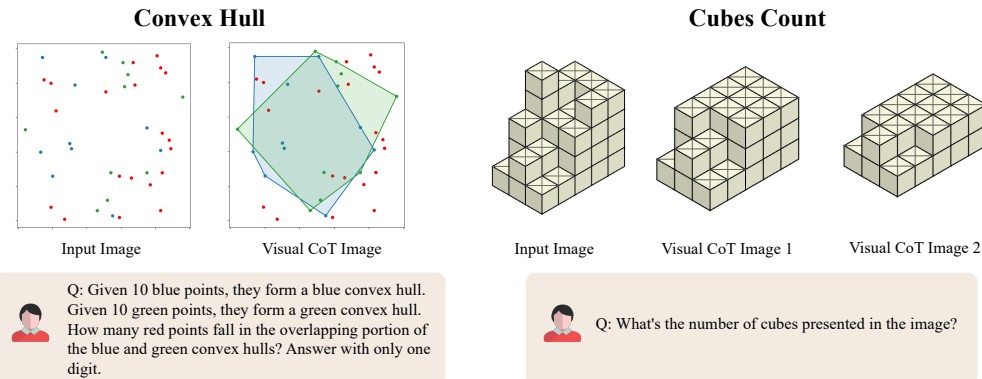

**Convex Hull**

Input Image · Visual CoT Image

Q: Given 10 blue points, they form a blue convex hull. Given 10 green points, they form a green convex hull. How many red points fall in the overlapping portion of the blue and green convex hulls? Answer with only one digit.

**Cubes Count**

Input Image · Visual CoT Image 1 · Visual CoT Image 2

Q: What's the number of cubes presented in the image?

Figure 5: Illustrative cases for Convex Hull task (left) and Cubes Count task (right).

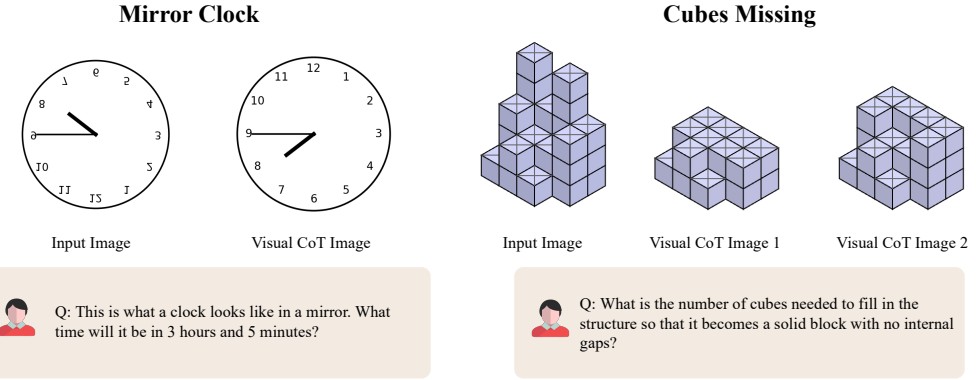

**Mirror Clock**

Input Image · Visual CoT Image

Q: This is what a clock looks like in a mirror. What time will it be in 3 hours and 5 minutes?

**Cubes Missing**

Input Image · Visual CoT Image 1 · Visual CoT Image 2

Q: What is the number of cubes needed to fill in the structure so that it becomes a solid block with no internal gaps?

Figure 6: Illustrative cases for Mirror Clock task (left) and Cubes Missing task (right).

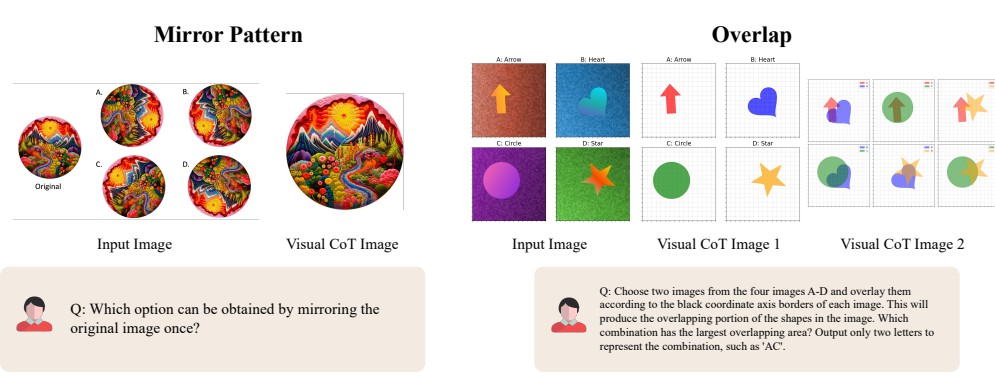

Figure 7: Illustrative cases for Mirror Pattern task (left) and Overlap task (right).

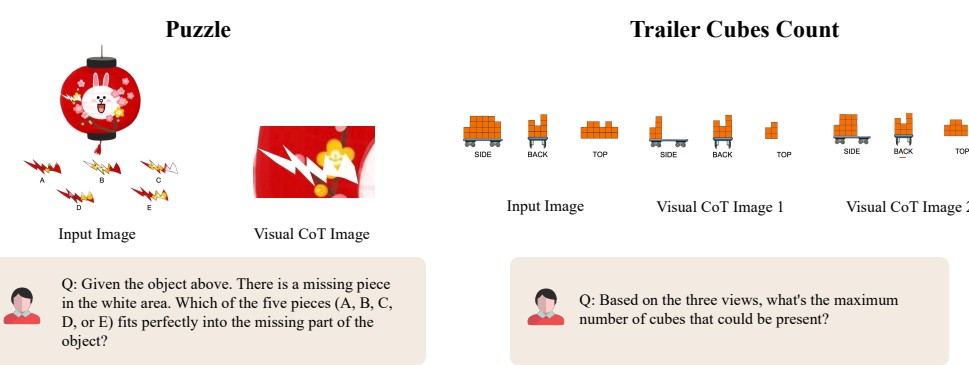

Figure 8: Illustrative cases for Puzzle task (left) and Trailer Cubes Count task (right).

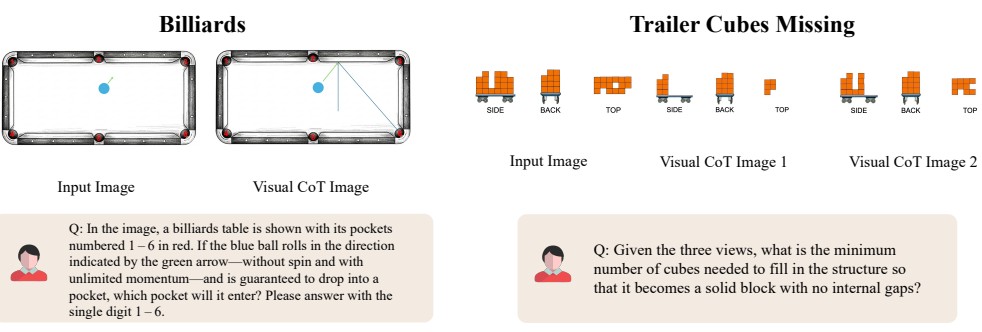

Figure 9: Illustrative cases for Puzzle task (left) and Trailer Cubes Count task (right).

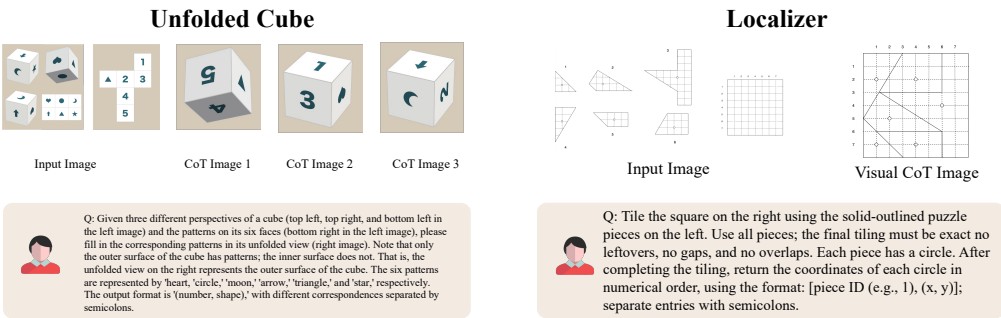

Figure 10: Illustrative cases for Unfolded Cube task (left) and Localizer task (right).

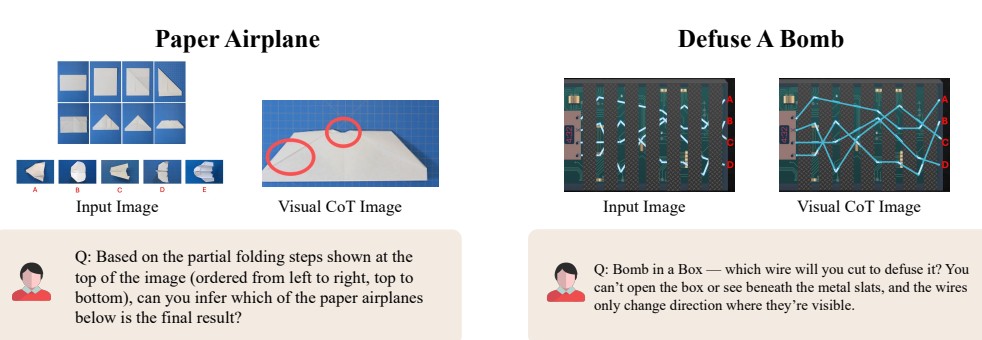

Figure 11: Illustrative cases for Paper Airplane task (left) and Defuse A Bomb task (right).

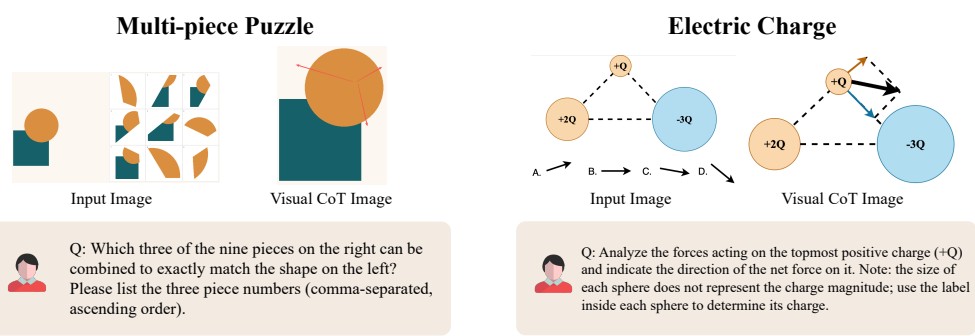

Figure 12: Illustrative cases for Multi-piece Puzzle task (left) and Electric Charge task (right).

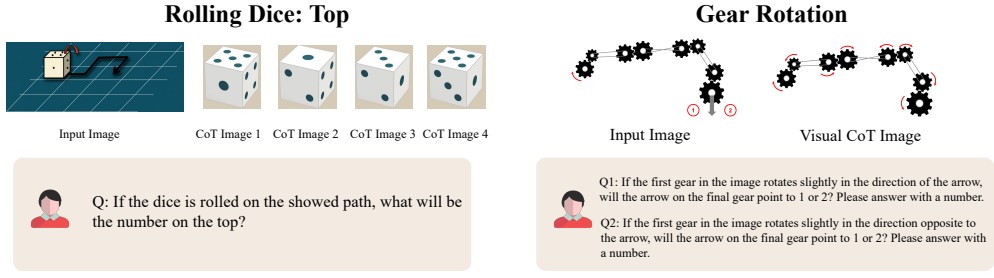

Figure 13: Illustrative cases for Rolling Dice: Top task (left) and Gear Rotation task (right).

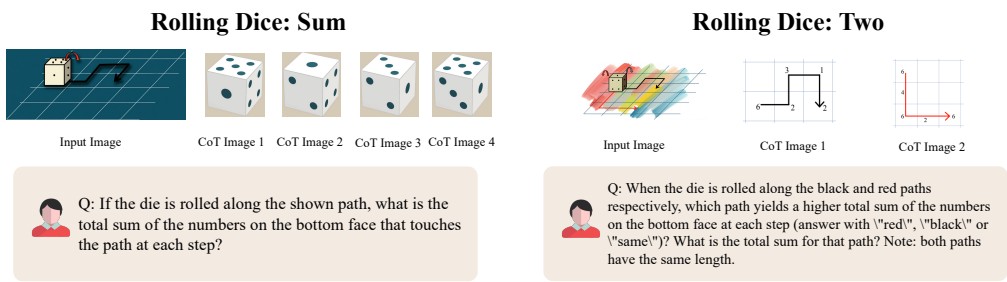

Figure 14: Illustrative cases for Rolling Dice: Sum task (left) and Rolling Dice: Two task (right).

