# OpenReview forum: "When Visualizing is the First Step to Reasoning: MIRA, a Benchmark for Visual Chain-of-Thought"
_ICLR.cc/2026/Conference — ICLR 2026 Conference Withdrawn Submission_

### Official Review · Reviewer_XjS7 · 2025-10-25

**Soundness:** 3
**Presentation:** 3
**Contribution:** 3
**Rating:** 4
**Confidence:** 4

**Summary:**

This paper construct a benchmark for evaluating multimodal problems reasoning performance. It measures the performance of MLLMs  under both text-only CoT and Visual-CoT input settings.

For text-only CoT, the evaluation includes two modes: one using a generic system prompt, and another using a task-specific system prompt tailored to the specific question. For Visual-CoT, the evaluation uses human-prepared images as the visual chain of thought. Under all evaluation settings, the MLLMs show limited performance.

**Strengths:**

1. This paper introduces high-quality benchmark with human labeling and inspection. The benchmark spans 20 task types and includes 546 carefully designed examples. On this benchmark, current models perform poorly, indicating that it presents a challenging set of tasks for existing models.

2. By providing well-designed text CoTs and visual CoTs, the authors tested the upper performance of current models on this task. Even with high-quality simulated Visual-CoT reasoning (manually annotated intermediate images), the models still struggled to answer these questions effectively. These simulated Visual-CoT reasoning traces can serve as a foundation for future research on enabling models to autonomously generate their own internal Visual CoTs.

**Weaknesses:**

1. As a benchmark-oriented study, and the process does not involve automated construction by models, it may not align closely with ICLR’s primary interests. It would likely be more suitable for the data track.

2. In addition to providing text-CoT prompts and simulated visual CoTs for questions to test the model’s upper-bound performance under high-quality CoT conditions, there doesn’t seem to be a fundamental difference between this work and other multi-modality reasoning benchmarks, such as [1]. Of course, [1] was released within about a month of the ICLR submission deadline, it can be considered a contemporaneous work. I still expect the authors to include a more detailed discussion comparing their approach with similar works in this area.


3. I suggest that the authors include the CoT construction process in the main text, as this is crucial for understanding the experiments. Especially the fact that the simulated visual CoTs are human-constructed. This raises a question: these human-constructed CoTs might function more like visual hints than genuine model reasoning traces, have the authors considered comparing their benchmark results with studies where models autonomously generate visual CoTs? For instance, see [2].

[1] Li, C., Wu, W., Zhang, H., Li, Q., Gao, Z., Xia, Y., ... & Wei, F. (2025). 11plus-bench: Demystifying multimodal LLM spatial reasoning with cognitive-inspired analysis. arXiv preprint arXiv:2508.20068.

[2] Li, C., Wu, W., Zhang, H., Xia, Y., Mao, S., Dong, L., Vulić, I., & Wei, F. (2025). Imagine While Reasoning in Space: Multimodal Visualization-of-Thought. In Forty-second International Conference on Machine Learning (ICML).

**Questions:**

Please check the comments in Weaknesses section.

---

### Official Review · Reviewer_xQL5 · 2025-10-26

**Soundness:** 2
**Presentation:** 3
**Contribution:** 2
**Rating:** 2
**Confidence:** 4

**Summary:**

This paper introduces MIRA, a novel benchmark designed to evaluate the visual reasoning capabilities of Multimodal Large Language Models (MLLMs). The authors argue that current Chain-of-Thought (CoT) prompting methods are limited to the textual domain and fail on tasks that are intrinsically visual, where humans often resort to sketching or visualizing intermediate steps. MIRA consists of 546 challenging problems across four domains (Geometry, Physics, Puzzles, and Causal Transformations) that require such intermediate visual reasoning. Experiments on a wide range of state-of-the-art MLLMs show that MIRA is highly challenging, underscoring the limitations of current models and the importance of this reasoning modality.

**Strengths:**

1. The proposed three-level evaluation protocol is a major strength. By systematically comparing Direct Evaluation, Text-CoT, and Simulated Visual-CoT, it provides clear evidence for their central claims.

2. The authors have conducted an extensive evaluation across a large and representative set of MLLMs, including top-tier closed-source models and various open-weight alternatives.

**Weaknesses:**

1. **Limited Scale and Generality of the Benchmark**: The size of this benchmark (546 examples) is relatively modest, which could limit the statistical power of the conclusions. Besides, it remains an open question how well the findings would generalize to more common, real-world visual reasoning scenarios that are less puzzle-like.

2. **Lack of a Random Baseline**: The paper does not provide a random-chance baseline for its tasks.With reported accuracies in most tasks under 20%,  it is difficult to ascertain whether the models are exhibiting any meaningful reasoning or simply performing near (or even below) random guessing.

3. **Uniformly High Task Difficulty Limits Diagnostic Granularity**: The consistently low performance across nearly all models raises a question about the benchmark's difficulty calibration. While demonstrating the limitations of current SOTA models is a valid goal, the tasks may be uniformly too difficult to serve as a fine-grained diagnostic tool. A benchmark that includes a spectrum of difficulty levels (e.g., easy, medium, hard) would be more valuable. It would allow researchers to identify the "breaking point" of different models and architectures, revealing partial capabilities rather than just wholesale failure.

**Questions:**

Given that the performance of even the strongest models is quite low across most tasks, have the authors considered establishing a human performance baseline for the MIRA benchmark? This data would be valuable for calibrating the absolute difficulty of the tasks and providing a clear "upper bound" to better contextualize the models' scores and quantify the gap to human-level reasoning.

---

### Official Review · Reviewer_RL6B · 2025-10-26

**Soundness:** 2
**Presentation:** 2
**Contribution:** 2
**Rating:** 2
**Confidence:** 5

**Summary:**

This paper introduces MIRA, a new benchmark designed to evaluate the reasoning capabilities of MLLMs in tasks where generating intermediate visual steps is essential. The authors argue that standard textual CoT prompting is insufficient for problems requiring spatial or geometric reasoning, which humans often solve by "drawing to think." The MIRA benchmark includes 546 problems annotated with gold-standard, step-by-step visual intermediate states to facilitate a novel "Visual-CoT" evaluation. Experiments on state-of-the-art models show they perform poorly with standard inputs and textual CoT prompting. However, when models are provided with the intermediate visual cues in the Visual-CoT setting, their performance consistently and significantly improves, highlighting the current limitations of text-only reasoning and the critical role of visual CoT.

**Strengths:**

1. The paper proposes an interesting scenario that requires the generation of intermediate visual images to solve reasoning tasks.

2. The curated intermediate CoT images provided in the dataset are of high quality.

**Weaknesses:**

1. **Inconsistent Answer Formats:** The benchmark employs a wide variety of answer formats, including multiple-choice, free-form text, numeric values, lists, and even custom coordinate-based formats (e.g., the localizer task). This inconsistency makes it difficult to perform robust statistical analysis or establish a consistent "random guess" baseline for comparison across the diverse task types.

2. **Unclear Narrative and Analysis:** The paper's core objective is not clearly articulated; it remains ambiguous about the specific MLLM ability MIRA is intended to measure. The manuscript dedicates excessive space to discussing the limitations of textual CoT and contrasting it with Visual-CoT, which detracts from the benchmark's own merits. This is compounded by a lack of fine-grained error analysis; the paper seems to attribute the models' poor performance primarily to the failures of textual CoT without a deeper investigation. It doesn't analyze *why* models fail on the visual tasks themselves (e.g., perception, spatial misinterpretation, logical error, failure to track state).

3. **Limited Scale and Discriminative Power:** The dataset scale is small. As a result, the reported results across various models are not sufficiently discriminative. Most models perform similarly poorly, making it difficult to use the benchmark to rank or meaningfully distinguish their relative reasoning capabilities.

4.  **Visual-CoT Setting Misalignment:** The Visual-CoT setting does not effectively evaluate a reasoning trajectory. By providing the gold-standard intermediate images as inputs, the benchmark does not evaluate the model's reasoning trajectory. Instead, it reframes the task as a multi-image reasoning problem where the intermediate images act as "hints" rather than as a "thought process" complementary to the model's own reasoning. This setup fails to capture the dynamic, interleaved nature of a true visual reasoning process. A more effective evaluation might involve providing these visual cues in an interactive, simulative manner, rather than as a single, complete set of inputs.

**Questions:**

See above

---

### Official Review · Reviewer_zfSt · 2025-10-31

**Soundness:** 3
**Presentation:** 3
**Contribution:** 3
**Rating:** 6
**Confidence:** 3

**Summary:**

The authors introduce MIRA, a new benchmark for model evaluation where reasoning across modalities -- i.e., in text and images -- helps solve the given tasks. They create a pipeline for generating this data, involving complexity-conditioned programmatic and manual generation with manual verification. They evaluate three different settings: direct evaluation, text-only chain of thought, and 'simulated visual-CoT reasoning' (giving the MLLM the correct intermediate drawing that it would ideally make, but that current models are apparently not capable of making).

**Strengths:**

- The paper successfully identifies shortcomings in existing datasets and makes a convincing argument for why it is needed.
- The pipeline for data generation and for evaluation are set up well, providing some degree of trust in the dataset.
- The results on the benchmark suggest new directions for the field to explore, highlighting types of questions that remain unsolved even by strong closed models.

**Weaknesses:**

- Tool-augmented methods are mentioned as part of the motivation (page 2) but not evaluated on the benchmark as far as I can tell. It would be interesting to see these results.
- 4.2 suggests Visual-CoT data may be a way forward with these types of questions, but these need to be manually created if I understand correctly. In what contexts would they actually be helpful given new problems won't have them at inference time, or will need to be annotated for that specifically? When is it easier for the human to provide the Visual-CoT annotation than just the final answer itself?
- Additional details such as the breakdown of problem sources and how many are manual vs not would be good to have known.
- Some additional references for intermediate visual reasoning: Zebra-CoT (Li et al), Whiteboard-of-Thought (Menon et al), SketchAgent (Vinker et al).

**Questions:**

- The motivation of simulated Visual-CoT makes sense to me as an evaluation setting, but how could it be actually applied to new problems as the authors allude to in 4.2?
- What is human performance on the benchmark?
- Are any methods that attempt to create intermediate visual representations evaluated other than the two unified MLLMs Bagel and Janus-Pro, which can't produce images on the way to text answers?

---

### Note · Authors · 2025-11-13

I have read and agree with the venue's withdrawal policy on behalf of myself and my co-authors.